# Pseudo-Mallows for Efficient Probabilistic Preference Learning

Qinghua Liu [1]   Valeria Vitelli [2]   Carlo Mannino [3]   Arnoldo Frigessi [2 4]   Ida Scheel [1]

## Abstract

We propose the Pseudo-Mallows distribution over the set of all permutations of $n$ items, to approximate the posterior distribution of the Bayesian Mallows model. The Bayesian Mallows model has been successfully used for recommender systems to learn personal preferences from highly incomplete users data. However current inference algorithms do not scale, preventing its use in real-time applications. The Pseudo-Mallows distribution is a product of univariate discrete Mallows-like distributions, where the quality of the approximation depends on the order of the $n$ items in the factorization sequence. In a variational setting, we optimize the variational order parameter by minimising a marginalized KL-divergence, conjecturing a certain form of the optimal variational order that depends on the data, and proposing an approximation algorithm for this discrete optimization. Empirical evidence and some theory support our conjecture. We demonstrate on clicking data that variational inference via the Pseudo-Mallows distribution allows much faster probabilistic preference learning compared to alternative MCMC-based options.

## 1. Introduction

Preference learning consists of estimating the unknown permutation of $n$ items that represents the comparative opinion about these items by an individual or user. Usually, the users have expressed only incomplete information about their ranking, such as pairwise preferences or clicks. When observing $N$ users who share preferences while also showing individual opinions, the aim is inferring on the shared consensus ranking of the items and on each individual complete ranking. Probabilistic preference learning recognises and quantifies the uncertainty of inference, by means of (posterior) distributions over the set $\mathcal{P}_n$ of all permutations of $n$ items. Probabilistic preference learning is therefore a key component of a recommender system, whose aim is to present to each user a selection of items based on his/her personal preference, and where uncertainty quantification is key to avoid spamming users.

The study of distributions over $\mathcal{P}_n$ started with the work by Diaconis (Diaconis, 1988). Recently, Shah (2021) collected several approaches to build probability distributions over permutations. Liu et al. (2019a) present additional options, in particular the Mallows model (Mallows, 1957). Difficulties originate from the dimension of $\mathcal{P}_n$ for larger $n$. In inference, the incompleteness of the data means that appropriate conditional distributions over $\mathcal{P}_n$ need to be constructed. A typical case for a recommender system could have hundreds or more of items, ten-thousands or even more users, each expressing preferences by clicking/liking some few items.

The use of the Mallows model in a Bayesian setting for probabilistic preference learning was first proposed in Vitelli et al. (2018). The Mallows is an exponential family distribution assuming that the latent individual ranking of the items is distributed around a consensus $\boldsymbol{\rho} \in \mathcal{P}_n$, with probability decreasing as a function of the distance to $\boldsymbol{\rho}$. Bayesian inference can be performed by augmentation and Markov Chain Monte Carlo (MCMC), which however mixes slowly for large numbers of items $n$ and users $N$, and faces difficulties in the multimodality of the discrete posterior distribution. A fast alternative to MCMC is Variational Inference (VI) (Blei et al., 2017; Zhang et al., 2018; Salimans et al., 2015), consisting in approximating the posterior $p(\boldsymbol{\theta}|\mathbf{D})$ for a parameter $\boldsymbol{\theta}$, given data $\mathbf{D}$, by a simpler distribution $q(\boldsymbol{\theta})$ belonging to a family of distributions $\mathcal{Q}$. The optimal variational approximation $q^*(\boldsymbol{\theta})$ for $p(\boldsymbol{\theta}|\mathbf{D})$ is selected by minimizing the Kullback-Leibler (KL) divergence (Kullback & Leibler, 1951) between $p(\boldsymbol{\theta}|\mathbf{D})$ and $q(\boldsymbol{\theta})$

$$q^*(\boldsymbol{\theta}) = \underset{q(\boldsymbol{\theta}) \in \mathcal{Q}}{\operatorname{argmin}} KL(q(\boldsymbol{\theta})||p(\boldsymbol{\theta}|\mathbf{D})). \qquad (1)$$

Typically, instead of minimizing the KL divergence directly, it is more convenient to maximize the evidence lower bound

[1]Dept. of Mathematics, University of Oslo, Oslo, Norway [2]OCBE, University of Oslo, Oslo, Norway [3]Mathematics and Cybernetics, Sintef Digital, Oslo, Norway [4]Department of Informatics, University of Oslo, Oslo, Norway. Correspondence to: Ida Scheel <idasch@math.uio.no>, Valeria Vitelli <valeria.vitelli@medisin.uio.no>.

*Proceedings of the 43$^{rd}$ International Conference on Machine Learning*, Seoul, South Korea. PMLR 306, 2026. Copyright 2026 by the author(s).

(ELBO) (Hoffman & Johnson, 2016). Algorithms that are commonly used to optimize the ELBO, e.g. CAVI (Bishop, 2006) or SVI (Hoffman et al., 2013) converge fast, but they are typically difficult for discrete distributions, and most VI approaches have focused on continuous exponential family distributions (Hensman et al., 2012; Wainwright & Jordan, 2008). Furthermore, VI is often built upon the mean-field assumption, i.e., when $q(\boldsymbol{\theta})$ is assumed factorizable into univariate distributions in the components of the vector $\boldsymbol{\theta}$ (Xing et al., 2012). The mean-field assumption makes it difficult to capture the dependencies between the various parameters (Wang & Titterington, 2005), an aspect that is essential for a ranking vector $\boldsymbol{\rho} \in \mathcal{P}_n$.

We propose the Pseudo-Mallows distribution as an efficient VI approximation to the Bayesian Mallows Model (BMM) posterior distribution, without following the mean-field construction. Our proposed distribution allows sampling in the space of permutations conditional on incomplete data, thus enabling learning from incomplete preferences as well as from clicking data. The Pseudo-Mallows distribution inherits most favourable traits of the BMM, with the advantage of drawing independent samples, which adds efficiency. We propose a marginalised version of the KL-divergence and derive its equivalent ELBO. The optimal variational parameter of the Pseudo-Mallows distribution is a permutation itself, which has an interesting counter-intuitive form that we conjecture to be theoretically valid. We support our conjecture i) by empirical evidence obtained in small dimensions, ii) through a new optimization algorithm that minimises the marginalised KL-divergence by transforming the problem into a bipartite matching problem, and iii) by valid theory in special cases. Through simulations we show that not only is our proposed Pseudo-Mallows method highly efficient compared to MCMC, but also that it facilitates accurate inference and personalized recommendations to be made in a timely manner. To our knowledge, our proposed method is the first to develop variational inference for a discrete complex distribution over a permutation space. The only comparable variational approximations for discrete data are tailored to different classes of problems such as Bipartite Matching (Volkovs & Zemel, 2012) or biological sequences (Bouchard-Côté & Jordan, 2010; Rao & Kirk, 2025). Alternative scalable approaches such as spectral methods (Liu et al., 2024) lack uncertainty quantification, and EM-based methods are slow (Eliseussen et al., 2023).

The BMM has been shown favourable to other probabilistic approaches to preference learning (Liu et al., 2019a). Furthermore, Liu et al. (2019b) show that BMM is able to make personalized recommendations with similar accuracy as the state-of-the-art matrix factorization method (CF), while also providing interpretable uncertainty estimation and achieving much higher level of diversity. Introducing PseudoMallows as an efficient approximation to the BMM is an important

contribution for making it useful in high-dimensional settings.

The paper is structured as follows. In Section 2, we give a detailed explanation of the Pseudo-Mallows method for full rankings, wherein we introduce the Pseudo-Mallows distribution, and our conjecture of the optimal variational parameter that achieves the best approximation to the Bayesian Mallows model. In Section 3, we introduce the Pseudo-Mallows method to learn individual rankings from partial rankings and make personalized recommendations from clicking data. The methodology is then supported by simulation and real data studies in Section 4. In Section 4.1, we demonstrate the Pseudo-Mallows' ability to make fast inference from full ranking data, and in Section 4.2, we test the Pseudo-Mallows on a real life clicking dataset to showcase its ability to make accurate and fast personalized recommendations given a time limitation. Finally, we collect some concluding comments in Section 5. Additional theoretical details and empirical results are given in Appendices A-E. Details of all implemented algorithms are reported in Appendix F. Proofs are given in Appendix G.

## 2. Pseudo-Mallows-based VI for targeting the BMM posterior

Suppose we have a group of $N$ users and each user has, independently of all other users, given a full ranking of $n$ items. We denote the rank vector of user $j$ as $\mathbf{R}^j = (R_1^j, \ldots, R_n^j)$, where $R_i^j = k$ indicates that user $j$ gives to item $i$ rank $k$, and hence $\mathbf{R}^j \in \mathcal{P}_n, \forall j = 1, ..., N$. Let also $\mathbf{R} = \{\mathbf{R}^1, \ldots, \mathbf{R}^N\}$. Assuming the users are homogenous in the sense that they share their preferences to some degree, the Mallows model assumes that their rankings are distributed around a consensus parameter $\boldsymbol{\rho}$, with a scale parameter $\alpha$. Given $\alpha$, and assuming a non-informative uniform prior for $\boldsymbol{\rho}$, the BMM posterior distribution of the consensus parameter $\boldsymbol{\rho}$ is

$$P(\boldsymbol{\rho}|\alpha, \mathbf{R}^1, \ldots, \mathbf{R}^N) = \frac{\exp\left\{-\frac{\alpha}{n}\sum_{j=1}^N d(\mathbf{R}^j, \boldsymbol{\rho})\right\}}{Z_n(\alpha, \mathbf{R}^1, \ldots, \mathbf{R}^N)}, \tag{2}$$

where $d(,)$ is a right-invariant distance (Diaconis, 1988) between two rankings and

$$Z_n(\alpha, \mathbf{R}^1, \ldots, \mathbf{R}^N) = \sum_{\mathbf{r} \in \mathcal{P}_n} \exp\left\{-\frac{\alpha}{n}\sum_{j=1}^N d(\mathbf{R}^j, \mathbf{r})\right\}$$

is the normalizing constant, which does not depend on $\boldsymbol{\rho}$ when the distance is right-invariant. $\mathcal{P}_n$ is the space of permutations of $n$ items. We use the right-invariant footrule distance, defined as

$$\boldsymbol{d}(\mathbf{R}^j, \boldsymbol{\rho}) = \sum_{i=1}^n d(R_i^j, \rho_i) = \sum_{i=1}^n |R_i^j - \rho_i|.$$

A more detailed description of the BMM is given in Appendix A.

With the aim to build a VI framework for BMM, we introduce a new probability distribution on the space of permutations, named the Pseudo-Mallows distribution, to approximate the BMM posterior distribution (2). In order to obtain independent samples of $\boldsymbol{\rho}$ conveniently, our new distribution is a product of $n$ conditional distributions, which can be easily sampled from.

### 2.1. The Pseudo-Mallows distribution

The Pseudo-Mallows samples one element of the consensus ranking vector $\boldsymbol{\rho}$ at a time, conditional on the elements of $\boldsymbol{\rho}$ that have already been sampled, to ensure a valid ranking. The ordering $\boldsymbol{o} = (o_1, ..., o_n) \in \mathcal{P}_n$ in which the elements of $\boldsymbol{\rho}$ are being sampled is an essential part of the distribution. Given this ordering, which will be given a distribution and sampled as shown later, the Pseudo-Mallows sampling scheme is as follows for the first and $k$-th element of $\boldsymbol{\rho}$

$$q(\rho_{o_1}|\alpha, o_1, R_{o_1}^1, ..., R_{o_1}^N)$$

$$= \frac{\exp\{-\frac{\alpha}{n}\sum_{j=1}^N d(R_{o_1}^j, \rho_{o_1})\}}{\sum_{r\in\{1,..,n\}} \exp\{-\frac{\alpha}{n}\sum_{j=1}^N d(R_{o_1}^j, r)\}} \mathbb{1}_{\rho_{o_1}\in\{1,...,n\}},$$

$$q(\rho_{o_k}|\alpha, o_k, \rho_{o_1}, ..., \rho_{o_{k-1}}, R_{o_k}^1, ..., R_{o_k}^N) \qquad (3)$$

$$= \frac{\exp\{-\frac{\alpha}{n}\sum_{j=1}^N d(R_{o_k}^j, \rho_{o_k})\}}{\sum_{r\in\{1,..,n\}\setminus\{\rho_{o_1},...,\rho_{o_{k-1}}\}} \exp\{-\frac{\alpha}{n}\sum_{j=1}^N d(R_{o_k}^j, r)\}}$$

$$\cdot \mathbb{1}_{\rho_{o_k}\in\{1,..,n\}\setminus\{\rho_{o_1},...,\rho_{o_{k-1}}\}}, \text{ for } k = 2, ..., n$$

where $d(,)$ is the distance $d(R_i^j, \rho_i) = |R_i^j - \rho_i|$, which corresponds to the right-invariant footrule distance between two rankings $\mathbf{R}^j$ and $\boldsymbol{\rho}$, defined as $\sum_{i=1}^n d(R_i^j, \rho_i) = \sum_{i=1}^n |R_i^j - \rho_i|$.

Now, (3) corresponds to the following joint distribution for $\boldsymbol{\rho}$ conditional on the sampling ordering $\boldsymbol{o}$

$$q(\boldsymbol{\rho}|\alpha, \boldsymbol{o}, \mathbf{R})$$
$$= q(\rho_{o_1}|\alpha, o_1, R_{o_1}^1, ..., R_{o_1}^N) \cdot q(\rho_{o_2}|\alpha, o_2, \rho_{o_1} R_{o_2}^1, ..., R_{o_2}^N)$$
$$\cdots q(\rho_{o_{n-1}}|\alpha, o_{n-1}, \rho_{o_1}, ..., \rho_{o_{n-2}}, R_{n-1}^1, ..., R_{n-1}^N)$$
$$\cdot q(\rho_{o_n}|\alpha, o_n, \rho_{o_1}, ..., \rho_{o_{n-1}}, R_n^1, ..., R_n^N)$$
$$= \frac{\exp\left\{-\frac{\alpha}{n}\sum_{j=1}^N d(\mathbf{R}_j, \boldsymbol{\rho})\right\}}{Z_n^{PM}(\boldsymbol{\rho}, \alpha, \mathbf{R}, \boldsymbol{o})},$$
$$(4)$$

where $Z_n^{PM}(\boldsymbol{\rho}, \alpha, \mathbf{R}, \boldsymbol{o})$ indicates the product of all denominators in (3). Given an ordering $\boldsymbol{o}$, as well as $\alpha = \alpha^0$ and the full ranking data $\mathbf{R}$, we obtain $M$ samples from the Pseudo-Mallows distribution of $\boldsymbol{\rho}$ by sampling, for $m = 1, ..., M$, first $\rho_{o_1}^m$, followed by $\rho_{o_2}^m$ and so on, until lastly $\rho_{o_n}^m$, using the Multinomial distribution for $\rho_{o_k}$ given by (3). A full description of the algorithm can be seen in Algorithm 1 in Appendix F.

Equation (4) is a factorization in $n$ terms, each being a univariate distribution for one element of the consensus ranking vector. Note that the product of all terms on the numerator in (3) is effectively identical to the numerator of the Mallows distribution, which makes it plausible that the Pseudo-Mallows distribution shares similar characteristics of the Mallows distribution. The scale parameter $\alpha$ plays a similar role as in the Mallows distribution controlling the "peakness" of the distribution. Within each term on the numerator in (3), we measure the distance between $\rho_{o_k}$ and the rankings of item $o_k$ given by the users. In the denominator, we sum over all possible values that $\rho_{o_k}$ can still take, which implies that the ordering $\boldsymbol{o}$ in which the elements of $\boldsymbol{\rho}$ are sampled matters. This ordering is in fact the variational parameter to be optimized. An equivalent reparametrization can be used, for convenience, based on the inverse ranking corresponding to such an ordering. The inverse operator maps rankings into orderings (and viceversa), and is defined as $\mathcal{I} : \mathcal{P}_n \to \mathcal{P}_n$, where $\boldsymbol{v} = \mathcal{I}(\boldsymbol{o})$ with $\boldsymbol{v} = (v_1, ..., v_n)$ being the ranking such that $v_{o_k} = k$ (note that, when the items are conventionally labelled as the first $n$ integers, by definition also $o_{v_k} = k$). It is more convenient to assume that this ranking (and not the corresponding ordering $\boldsymbol{o} = \mathcal{I}(\boldsymbol{v})$) follows a distribution $g(\boldsymbol{v})$ defined on $\mathcal{P}_n$. By construction, using the multiplication rule of distributions and appropriate conditional independence, and rewriting $q(\boldsymbol{\rho}|\alpha, \boldsymbol{o}, \mathbf{R}) = q(\boldsymbol{\rho}|\alpha, \boldsymbol{v}, \mathbf{R})$, the Pseudo-Mallows distribution defined as

$$Q(\boldsymbol{\rho}|\alpha, \mathbf{R}) = \sum_{\boldsymbol{v}\in\mathcal{P}_n} q(\boldsymbol{\rho}|\alpha, \boldsymbol{v}, \mathbf{R})g(\boldsymbol{v}) \qquad (5)$$

is a proper distribution on $\mathcal{P}_n$. This completes the definition of the Pseudo-Mallows distribution. The distribution $g(\boldsymbol{v})$ can be any proper distribution on the permutation space.

### 2.2. Optimal construction for the Pseudo-Mallows distribution

To obtain the best approximation to the Mallows posterior, we seek the distribution $g(\boldsymbol{v})$ that minimizes the Kullback-Leibler (KL) divergence between the Mallows posterior (2) and the Pseudo-Mallows distribution (5). To simplify the problem, we first consider the special case where $g(\boldsymbol{v})$ is a $\delta$-distribution with mass 1 on one ranking $\boldsymbol{v}$, which translates this optimization problem to searching for a ranking $\boldsymbol{v} \in \mathcal{P}_n$ that minimizes the KL-divergence, because it simplifies (5)

to $Q(\boldsymbol{\rho}|\alpha, \mathbf{R}) = q(\boldsymbol{\rho}|\alpha, \boldsymbol{v}, \mathbf{R})$. We can hence write the objective function as

$$\underset{\boldsymbol{v} \in \mathcal{P}_n}{\operatorname{argmin}} KL\Big(q(\boldsymbol{\rho}|\alpha, \mathbf{R}, \boldsymbol{v})||P(\boldsymbol{\rho}|\alpha, \mathbf{R})\Big), \qquad (6)$$

or equivalently,

$$\underset{\mathbf{o} \in \mathcal{P}_n}{\operatorname{argmin}} KL\left(q(\boldsymbol{\rho}|\alpha, \mathbf{R}, \mathbf{o})||P(\boldsymbol{\rho}|\alpha, \mathbf{R})\right) \qquad (7)$$

for the ordering $\boldsymbol{o} = \mathcal{I}(\boldsymbol{v})$. As usual in variational inference, instead of minimizing the KL divergence, it is convenient to maximize the Evidence Lower Bound (ELBO) $\mathcal{L}$, defined as the lower bound on the log marginal probability of the data (Zhang et al., 2018). We now derive the ELBO expression for the Pseudo-Mallows approximation of the Mallows posterior distribution.

We can write the KL divergence in equation (7) as

$$KL\left(q^{\mathbf{o}}(\boldsymbol{\rho}|\text{data})||P(\boldsymbol{\rho}|\text{data})\right)$$
$$= \sum_{\boldsymbol{\rho} \in \mathcal{P}_n}\left[\log\left(\frac{q^{\mathbf{o}}(\boldsymbol{\rho}|\text{data})}{P(\boldsymbol{\rho}|\text{data})}\right) \cdot q^{\mathbf{o}}(\boldsymbol{\rho}|\text{data})\right]$$

where for the ease of notation we simply write $P(\boldsymbol{\rho}|\text{data})$ for $P(\boldsymbol{\rho}|\alpha, \mathbf{R}^1, \ldots, \mathbf{R}^N)$, and $q^{\mathbf{o}}(\boldsymbol{\rho}|\text{data})$ for $q(\boldsymbol{\rho}|\alpha, \mathbf{R}^1, \ldots, \mathbf{R}^N, \mathbf{o})$. Using (2) and (4), this translates to

$$KL\left(q^{\mathbf{o}}(\boldsymbol{\rho}|\text{data})| |P(\boldsymbol{\rho}|\text{data})\right) =$$

$$\sum_{\boldsymbol{\rho} \in \mathcal{P}_n}\left[\log\left(\frac{\exp\{-\frac{\alpha}{n}\sum_{j=1}^{N}d(\mathbf{R}^j, \boldsymbol{\rho})\}}{Z_n^{PL}(\boldsymbol{\rho}, \alpha, \mathbf{R}^1, \ldots, \mathbf{R}^N, \mathbf{o})}\right.\right.$$

$$\left.\cdot \frac{Z_n(\alpha, \mathbf{R}^1, \ldots, \mathbf{R}^N)}{\exp\{-\frac{\alpha}{n}\sum_{j=1}^{N}d(\mathbf{R}^j, \boldsymbol{\rho})\}}\right) \cdot \frac{\exp\{-\frac{\alpha}{n}\sum_{j=1}^{N}d(\mathbf{R}^j, \boldsymbol{\rho})\}}{Z_n^{PL}(\boldsymbol{\rho}, \alpha, \mathbf{R}^1, \ldots, \mathbf{R}^N, \mathbf{o})}\right].$$

This expression can be re-written as

$$KL\left(q^{\mathbf{o}}(\boldsymbol{\rho}|\text{data}) ||P(\boldsymbol{\rho}|\text{data})\right)$$

$$= \sum_{\boldsymbol{\rho} \in \mathcal{P}_n}\left[\log(Z_n(\alpha, \mathbf{R}^1, \ldots, \mathbf{R}^N))\right.$$

$$\left.\cdot \frac{\exp\{-\frac{\alpha}{n}\sum_{j=1}^{N}d(\mathbf{R}^j, \boldsymbol{\rho})\}}{Z_n^{PL}(\boldsymbol{\rho}, \alpha, \mathbf{R}^1, \ldots, \mathbf{R}^N, \mathbf{o})}\right]$$

$$- \sum_{\boldsymbol{\rho} \in \mathcal{P}_n}\left[\log(Z_n^{PL}(\boldsymbol{\rho}, \alpha, \mathbf{R}^1, \ldots, \mathbf{R}^N, \mathbf{o}))\right.$$

$$\left.\cdot \frac{\exp\{-\frac{\alpha}{n}\sum_{j=1}^{N}d(\mathbf{R}^j, \boldsymbol{\rho})\}}{Z_n^{PL}(\boldsymbol{\rho}, \alpha, \mathbf{R}^1, \ldots, \mathbf{R}^N, \mathbf{o})}\right].$$

We observe that the term $\log(Z_n(\alpha, \mathbf{R}_1, \ldots, \mathbf{R}_N))$ does not depend on $\boldsymbol{\rho}$ thanks to right-invariance of the distance. Therefore it can be taken out of the summation, leaving us with a summation of the Pseudo-Mallows posterior over all possible permutations in $\mathcal{P}_n$, which equals 1.

We thus obtain the ELBO as

$$\text{ELBO} = \sum_{\boldsymbol{\rho} \in \mathcal{P}_n}\left[\frac{\exp\left\{-\frac{\alpha}{n}\sum_{j=1}^{N}d(\mathbf{R}^j, \boldsymbol{\rho})\right\}}{\frac{Z_n^{PM}(\boldsymbol{\rho}, \alpha, \mathbf{R}, \mathbf{o})}{\log(Z_n^{PM}(\boldsymbol{\rho}, \alpha, \mathbf{R}, \mathbf{o}))}}\right], \qquad (8)$$

which we need to maximize in $\boldsymbol{o} \in \mathcal{P}_n$ to minimize (7).

In order to complete the optimal Pseudo-Mallows construction, we need to first define the V-rankings and the $\mathcal{V}$-set of a permutation $\boldsymbol{r}$.

**Definition 2.1.** Fix a permutation $\boldsymbol{r} \in \mathcal{P}_n$. Define $\boldsymbol{x} = \mathcal{I}(\boldsymbol{r})$, i.e. such that $r_{x_i} = i$, for $i = 1, ..., n$. Let $n = 2m-1$ if $n$ is odd, and $n = 2m$ if $n$ is even. The $\mathcal{V}$-set of $\boldsymbol{r}$, called $\mathcal{V}_{\boldsymbol{r}}$, is the set of permutations determined by $\boldsymbol{r}$ in the following way:

If $n$ is odd: $\mathcal{V}_{\boldsymbol{r}} = \{\boldsymbol{v} \in \mathcal{P}_n : v_{x_m} = 1; (v_{x_{m-k}}, v_{x_{m+k}}) = (2k, 2k+1)$ or $(2k+1, 2k), k = 1, ..., m-1\}$.
If $n$ is even, $\mathcal{V}_{\boldsymbol{r}} = \{\boldsymbol{v} \in \mathcal{P}_n : (v_{x_{m-k}}, v_{x_{m+k+1}}) = (2k+1, 2k+2)$ or $(2k+2, 2k+1), k = 0, ..., m-1\}$.

Any permutation that belongs to the $\mathcal{V}$-set of $\boldsymbol{r}$ is called a V-ranking of $\mathcal{V}_{\boldsymbol{r}}$.

Given a permutation of $n$ items $\boldsymbol{r}$, for $n$ odd, $\mathcal{V}_{\boldsymbol{r}}$ is the set of rankings such that the middle ranked item in $\boldsymbol{r}$ is top-ranked and the top- and bottom-ranked items in $\boldsymbol{r}$ are bottom-ranked. For $\boldsymbol{x} = \mathcal{I}(\boldsymbol{r})$, we define $\mathcal{V}_{\boldsymbol{x}}$ as the set of orderings corresponding to the permutations in $\mathcal{V}_{\boldsymbol{r}}$. With this definition we propose the optimal construction of the Pseudo-Mallows. An extensive theoretical and empirical investigation supporting the conjecture is reported in Appendix B.

**Optimal Pseudo-Mallows Conjecture.** Assume $\mathbf{R}^j \sim$ Mallows $(\boldsymbol{\rho}^0, \alpha), j = 1, ...., N$. For any ordering $\boldsymbol{o}^* \in \mathcal{V}_{\boldsymbol{x}^0}$ where $\boldsymbol{x}^0 = \mathcal{I}(\boldsymbol{\rho}^0)$, we conjecture that

$$\boldsymbol{o}^* = \underset{\boldsymbol{o} \in \mathcal{P}_n}{\operatorname{argmax}} \lim_{N \to \infty} \sum_{\boldsymbol{\rho} \in \mathcal{P}_n}\left[\frac{\exp\left\{-\frac{\alpha}{n}\sum_{j=1}^{N}d(\mathbf{R}^j, \boldsymbol{\rho})\right\}}{\frac{Z_n^{PM}(\boldsymbol{\rho}, \alpha, \mathbf{R}, \mathbf{o})}{\log(Z_n^{PM}(\boldsymbol{\rho}, \alpha, \mathbf{R}, \mathbf{o}))}}\right].$$

In other words, we first compute the $\mathcal{V}$-set based on $\boldsymbol{x}^0$. We conjecture that any ordering $\boldsymbol{o}$ that belongs to $\mathcal{V}_{\boldsymbol{x}^0}$ is the optimal ordering that maximizes the ELBO, as described in Equation (8), when $N$ diverges. For the corresponding ranking $\boldsymbol{v} = \mathcal{I}(\boldsymbol{o})$, this means that the all V-rankings that belong to $\mathcal{V}_{\boldsymbol{\rho}^0}$ maximize the ELBO. Some intuition supporting this conjecture is given in Appendix B, as well as an empirical study showing that all V-rankings appear to be equivalently

effective at achieving a good approximation of the Mallows posterior.

From the conjecture it can be deduced that rather than making the simplified assumption that $g(\boldsymbol{v})$ is a $\delta$-distribution, it is more natural to assume $g(\boldsymbol{v})$ to be a Uniform distribution on all V-rankings that belong to $\mathcal{V}_{\boldsymbol{\rho}^0}$. Then the Pseudo-Mallows distribution can be expressed as follows

$$Q(\boldsymbol{\rho}|\alpha, \mathbf{R}) = \sum_{\boldsymbol{v} \in \mathcal{P}_n} q(\boldsymbol{\rho}|\alpha, \mathbf{R}, \boldsymbol{v}) g^*(\boldsymbol{v}), \qquad (9)$$

where

$$g^*(\boldsymbol{v}|\mathcal{V}_{\boldsymbol{\rho}^0}) = \begin{cases} \frac{1}{|\mathcal{V}_{\boldsymbol{\rho}^0}|} & \text{if } \boldsymbol{v} \in \mathcal{V}_{\boldsymbol{\rho}^0} \\ 0 & \text{otherwise.} \end{cases} \qquad (10)$$

### 2.3. Inferring the $\mathcal{V}$-set from data

#### 2.3.1. THE ASYMPTOTIC $N \to \infty$ CASE

Since $\boldsymbol{\rho}^0$ is unknown in practice, we need to infer $\mathcal{V}_{\boldsymbol{\rho}^0}$ from data.

**Lemma 2.2.** *Assume* $\mathbf{R} \in \mathcal{P}_n$ *and* $\mathbf{R} \sim Mallows(\boldsymbol{\rho}^0, \alpha)$. *Let* $o_i^0$ *be such that* $\rho_{o_i^0}^0 = i$, *for* $i = 1, ..., n$. *For a fixed* $\boldsymbol{\rho}^0$ *and* $\alpha \in (0, \infty)$, *it holds that* $\mathbb{E}[R_{o_j^0}|\boldsymbol{\rho}^0, \alpha] < \mathbb{E}[R_{o_{j+1}^0}|\boldsymbol{\rho}^0, \alpha]$.

See Appendix G.3 for a proof. We now define the rank operator:

**Definition 2.3.** The rank vector of a real number vector $(y_1, ..., y_n)$, is defined by:

$$rank(y_1, ..., y_n) = (r_1, ..., r_n), \text{ s.t. } r_i = \sum_{j=1}^n \delta(y_i - y_j) \text{ for}$$

$i = 1, ..., n$, where $\delta(x) = \begin{cases} 1, & \text{if } x \geq 0 \\ 0, & \text{if } x < 0 \end{cases}$.

Assuming the ranking data are realizations from the Mallows distribution, the following theorem follows directly from **Lemma** 2.2 and from the fact that $\frac{1}{N}\sum_{j=1}^N R_i^j \to \mathbb{E}[R_i|\boldsymbol{\rho}^0, \alpha]$ almost surely as $N \to \infty$, $\forall i = 1, ..., n$, by the strong law of large numbers:

**Theorem 2.4.** *As* $N \to \infty$, *and* $\forall \alpha > 0$,

$$rank(\frac{1}{N}\sum_{j=0}^N R_1^j, ..., \frac{1}{N}\sum_{j=0}^N R_n^j) \to$$
$$rank(\mathbb{E}[R_1|\boldsymbol{\rho}^0, \alpha], ..., \mathbb{E}[R_n|\boldsymbol{\rho}^0, \alpha]) = \boldsymbol{\rho}^0.$$

In conclusion, for the $N \to \infty$ case, $\mathcal{V}_{\boldsymbol{\rho}^0}$ can be inferred from data by the $\mathcal{V}$ set of $rank(\frac{1}{N}\sum_{j=0}^N R_1^j, ..., \frac{1}{N}\sum_{j=0}^N R_n^j)$.

#### 2.3.2. THE CASE OF A FINITE NUMBER OF USERS

For finite $N$, $\boldsymbol{\rho}^0$ and therefore $\mathcal{V}_{\boldsymbol{\rho}^0}$ cannot be accurately inferred from the data. We define the estimate $\hat{\boldsymbol{\rho}}^0 = rank(\frac{1}{N}\sum_{j=0}^N R_1^j, ..., \frac{1}{N}\sum_{j=0}^N R_n^j)$ and its corresponding $\mathcal{V}$-set $\mathcal{V}_{\hat{\boldsymbol{\rho}}^0}$, which will differ from $\mathcal{V}_{\boldsymbol{\rho}^0}$. To robustify the procedure we thus introduce some variability around the set $\mathcal{V}_{\hat{\boldsymbol{\rho}}^0}$ to increase the precision of the Pseudo-Mallows approximation as compared to using the unknown true $\mathcal{V}_{\boldsymbol{\rho}^0}$. Instead of simply using a uniform distribution over the $\mathcal{V}$-set $\mathcal{V}_{\hat{\boldsymbol{\rho}}^0}$, i.e., $g^*(\boldsymbol{v}|\mathcal{V}_{\hat{\boldsymbol{\rho}}^0})$ as in (10), we sample from $g'(\boldsymbol{v}|\hat{\boldsymbol{\rho}}^0, \sigma)$ defined as follows

- $\hat{\boldsymbol{v}} \sim g^*(\hat{\boldsymbol{v}}|\mathcal{V}_{\hat{\boldsymbol{\rho}}^0})$

- $y_i \sim \mathcal{N}(y_i|\hat{v}_i, \sigma)$ for $i = 1, ..., n$

- $\boldsymbol{v} = rank(y_1, ..., y_n)$.

That is to say, we start from a ranking $\hat{\boldsymbol{v}}$ from the $\mathcal{V}$ set $\mathcal{V}_{\hat{\boldsymbol{\rho}}^0}$, and for each item $i$, we introduce a perturbation by drawing from a Gaussian distribution centred on $\hat{v}_i$ with some standard deviation $\sigma$. We then build a perturbed V-ranking using the rank operator. In this way, a smaller KL-divergence of the Pseudo-Mallows from the Mallows posterior can usually be achieved in the case when the estimate of $\boldsymbol{\rho}^0$ is inaccurate due to limited sample size. A study of the performance of this procedure is reported in Appendix C. It shows that the optimal $\sigma$ seems to depend on $\alpha$, $N$ and $n$. In particular, the optimal $\sigma$ decreases towards 0 as $\alpha$ and $N$ increase and $n$ decreases. Also, when $\alpha$ and $N$ are sufficiently large and/or $n$ is sufficiently small, the different choices of $\sigma$ do not affect the KL-divergence.

The Pseudo-Mallows algorithm for sampling $\boldsymbol{\rho}$ based on full ranking data is essentially Algorithm 1 extended by replacing a fixed sampling ordering by for each iteration sampling a perturbed V-ranking as described above, and inverting into an ordering. A more detailed description of the algorithm is summarized in Algorithm 2 in Appendix F.

### 2.4. Off-line estimation of $\alpha$

When given a ranking dataset where $\alpha$ is unknown, an estimate of $\alpha$ is required before the Pseudo-Mallows method can be used to make inference on the consensus parameter $\boldsymbol{\rho}$. In this section, we propose a method to estimate $\alpha$ when given a ranking dataset.

The scale parameter $\alpha$ describes the "peakness" of the Mallows distribution: a larger value of $\alpha$ indicates that the individual rankings are more closely distributed around the group consensus, while a smaller value of $\alpha$ indicates a flatter distribution. In other words, when $\alpha$ is large, the individual rankings are more similar to each other compared

to when $\alpha$ is smaller. We can measure the pairwise user similarity between two vectors $\mathbf{R}^j$ and $\mathbf{R}^k$ using cosine similarity, defined as

$$\text{sim}(\mathbf{R}^j, \mathbf{R}^k) = \frac{\mathbf{R}^j \cdot \mathbf{R}^k}{|\mathbf{R}^j| \cdot |\mathbf{R}^k|}.$$

To obtain a quick estimation of $\alpha$, we assume that the mean pairwise user cosine similarity of a ranking dataset resembles that of a simulated ranking dataset generated by drawing independent samples from the Mallows distribution with the same number of items $n$, and the same value of $\alpha$, and any chosen $\rho^0$. The mean pairwise user cosine similarity for $N$ users is defined as

$$Sim_{\alpha^0} = \frac{1}{N(N-1)} \sum_{j=1}^{N} \sum_{k \neq j} \text{sim}(\mathbf{R}^j, \mathbf{R}^k).$$

To find an approximation of $\alpha$, we first generate full ranking datasets of $N$ users and $n$ items by drawing from the Mallows distribution with a grid of $\alpha^0$ values, and a shared consensus parameter $\rho^0$, which is fixed as $(1, ..., n)$ for convenience. We then calculate the mean pairwise user cosine similarity of each simulated ranking dataset, and compare these with the mean pairwise user cosine similarity of the real data set to find the closest one, and use the corresponding $\alpha^0$ as estimate. Note that the number of users $N$ in each simulated dataset is unimportant, the only requirement for $N$ is that it should be large enough such that a good estimate of the mean pairwise user similarity can be obtained.

## 3. Clicking data

So far we have assumed to have full ranking data from all users, and the unknown parameter to infer has been the common consensus parameter $\boldsymbol{\rho}$. We now consider that we have only partial rankings from the users, specifically based indirectly on clicking data, where a click on items is interpreted as a preference for those items over the unclicked items. The full ranking is hence not known, only what items are assumed to be preferred (clicked) over others (unclicked). Bayesian inference with the Mallows model has already been extended to partial rankings (Vitelli et al., 2018) and clicking data (Liu et al., 2019a). Here we extend the Pseudo-Mallows distribution to partial rankings from clicking data and study the variational approach to inference in this case. The goal here is mainly to make inference on the full ranking $\mathbf{R}^j$ (which is only partially, indirectly observed) of each individual $j$, and use this to give recommendations on what to click next, based on what is inferred to be the preferences of the user.

### 3.1. Pseudo-Mallows for clicking data

Suppose user $j$ has clicked on a subset of the $n$ items $\mathcal{A} = (A_1, ..., A_n)$. For user $j$, let the binary vector

$\mathbf{B}^j = (b_1^j, ..., b_n^j)$ be defined by $b_i^j = 1$ if item $i$ is clicked and $b_i^j = 0$ otherwise. We also denote the set of items clicked by user $j$ as $\mathcal{A}^j = \{A_k : b_k^j = 1\}$, and the set of items not clicked by user $j$ as $\mathcal{A}^{j^c}$. Furthermore, we define $c_j = |\mathcal{A}^j|$ and $c_j^c := |\mathcal{A}^{j^c}| = n - c_j$.

Inference for the Mallows posterior in the clicking data case can be carried out through data augmentation. We first assume that each user $j$ has a latent ranking of all items $(R_1^j, ..., R_n^j)$ in mind, so that the user clicking behaviour is a consistent manifestation of that latent ranking. This consistency is ensured by assuming in practice that, for each user $j$, all clicked items are ranked higher than those not clicked, i.e., $R_i^j < R_k^j$, $\forall A_i \in \mathcal{A}^j$ and $A_k \in \mathcal{A}^{j^c}$. Within the groups of clicked or non-clicked items, no ordering is assumed. In this way, for each user $j$, $R_i^j \leq c_j$, $\forall i : A_i \in \mathcal{A}^j$ and $R_k^j \geq c_j + 1$, $\forall k : A_k \in \mathcal{A}^{j^c}$. We denote the set of rankings that are compatible with the binary clicks of user $j$ as $\mathcal{S}(\mathbf{B}^j) = \{\mathbf{R}^j \in \mathcal{P}_n : R_i^j \leq c_j, \forall i : A_i \in \mathcal{A}^j; R_k^j \geq c_j + 1, \forall k : A_k \in \mathcal{A}^{j^c}\}$. To make inference on $P(\boldsymbol{\rho}|\alpha, \mathbf{B}^1, ..., \mathbf{B}^N)$, the algorithm alternates between two steps: (i) estimate the full rankings for all users $\{\mathbf{R}^1, ..., \mathbf{R}^N\}$ based on the current estimation of $\boldsymbol{\rho}$ and on $\{\mathbf{B}^1, ..., \mathbf{B}^N\}$; (ii) update the estimate of $\boldsymbol{\rho}$ based on the current estimate of $\{\mathbf{R}^1, ..., \mathbf{R}^N\}$.

In step (ii), the sampling procedure is the same as when inferring for the group consensus $\boldsymbol{\rho}$ given full ranking data, as described in Section 2.

In step (i), we approximate the Mallows posterior

$$P(\mathbf{R}^j|\alpha, \boldsymbol{\rho}, \mathbf{B}^j) = P(\mathbf{R}^j|\alpha, \boldsymbol{\rho})\mathbb{1}_{\mathbf{R}^j \in \mathcal{S}(\mathbf{B}^j)} \quad (11)$$

by drawing a sample for each user $j$ independently from

$$Q^R(\mathbf{R}^j|\alpha, \boldsymbol{\rho}, \mathbf{B}^j) = \sum_{\boldsymbol{v}^j \in \mathcal{P}_n} f(\mathbf{R}^j|\alpha, \boldsymbol{\rho}, \mathbf{B}^j) \quad (12)$$

where $f(\cdot)$ is specified in the following. Given the assumption that all clicked items must be ranked higher than all unclicked items, $f(\cdot)$ factorizes into two independent, identical distributions, one for each group

$$f(\mathbf{R}^j|\alpha, \boldsymbol{\rho}, \mathbf{B}^j) = q^R(\mathbf{R}_{\mathcal{A}^j}^j|\alpha, \boldsymbol{\rho}^{t,j}, \boldsymbol{v}^{t,j}) \cdot g(\boldsymbol{v}^{t,j}) \\ \cdot q^R(\mathbf{R}_{\mathcal{A}^{j^c}}^j - c_j|\alpha, \boldsymbol{\rho}^{b,j}, \boldsymbol{v}^{b,j}) \cdot g(\boldsymbol{v}^{b,j}), \quad (13)$$

where $\boldsymbol{\rho}^{t,j} := \text{rank}(\boldsymbol{\rho}_{\mathcal{A}^j}) \in \mathcal{P}_{c_j}$, $\boldsymbol{\rho}^{b,j} := \text{rank}(\boldsymbol{\rho}_{\mathcal{A}^{j^c}}) \in \mathcal{P}_{c_j^c}$, $\boldsymbol{v}^{t,j} := (v_1^{t,j}, ..., v_{c_j}^{t,j}) \in \mathcal{P}_{c_j}$, and $\boldsymbol{v}^{b,j} := (v_1^{b,j}, ..., v_{c_j^c}^{b,j}) \in \mathcal{P}_{c_j^c}$ and $\boldsymbol{v}^j = (\boldsymbol{v}^{t,j}, \boldsymbol{v}^{b,j})$. Note that the input to $q^R(\cdot)$ for the unclicked items is the ranking vector $\mathbf{R}_{\mathcal{A}^{j^c}}^j$ shifted by $c_j$, which takes values in $\mathcal{P}_{c_j^c}$ after the shift. Here, $q^R(\cdot)$ follows the same type of item-by-item factorization form as $q(\cdot)$ in the Pseudo-Mallows for $\boldsymbol{\rho}$, and $g(\cdot)$ is some arbitrary distribution on the space of permutations defined on the

respective number of items (see Appendix D.2 for more details). Following the strategy for defining $q(\cdot)$ in the Pseudo-Mallows for $\boldsymbol{\rho}$, each term in $q^R(\cdot)$ can be factorized as shown in Appendix D.1.

In order to use (13) for performing inference on the users' individual rankings, we need to specify the user-specific distribution $g(\cdot)$ for the items ordering in the Pseudo-Mallows. We suggest to use the uniform distribution defined on $\mathcal{P}_{c_j}$ and $\mathcal{P}_{c_j^c}$, respectively, for the clicked and unclicked groups of items for each user $j$. Specifically,

$$
\begin{aligned}
g(\boldsymbol{v}) &= g(\boldsymbol{v}^{t,j})g(\boldsymbol{v}^{b,j}) \\
&= \begin{cases} \dfrac{1}{c_j!}\dfrac{1}{c_j^c!}, & \text{if } \boldsymbol{v}^{t,j} \in \mathcal{P}_{c_j} \text{ and } \boldsymbol{v}^{b,j} \in \mathcal{P}_{c_j^c} \\ 0, & \text{otherwise.} \end{cases}
\end{aligned}
$$

As clarified in Appendix D.2, this choice leads to an effective approximation of the Mallows posterior by means of the Pseudo- Mallows distribution in the augmentation step.

Similar to the full data case, an estimate of $\alpha$ is needed before the Pseudo-Mallows method can be applied to a clicking dataset to make inference on the group consensus parameter $\boldsymbol{\rho}$ and on the individual full ranking vector $\mathbf{R}^j$ for all users. A procedure for this is described in Appendix D.3.

### 3.2. Making personalized recommendations

We can now make personalized recommendations, as well as estimating the recommendation uncertainty for each user $j$, by the distribution of $j$'s full ranking $\mathbf{R}^j$ approximated from running Pseudo-Mallows for the clicking data. We continue with our assumption that for each user $j$, the $c_j$ clicked items are top ranked, and are not considered further for recommendation. In order to make $k$ recommendations for each user $j$, we need to infer the items that user $j$ would have ranked between $c_j + 1$ and $c_j + k$. For each user $j$, we can approximate for each item $i$ its probability to be ranked between $c_j + 1$ and $c_j + k$ given $\alpha$ and the clicking data $\mathbf{B}^1, ..., \mathbf{B}^N$, i.e.,

$$
P(c_j \le R_i^j \le c_j + k | \alpha, \mathbf{B}^1, ..., \mathbf{B}^N). \tag{14}
$$

The set of $k$ items with highest such probabilities can thus be recommended to the user.

## 4. Simulation and case study

### 4.1. Infer $\rho$ from full ranking data

In this section, we conduct experiments to compare computation speed and estimation accuracy for Pseudo-Mallows and Bayesian Mallows MCMC, when the task is to sample for the consensus parameter $\boldsymbol{\rho}$.

To conduct this experiment, we first generate full ranking datasets by sampling from the Mallows distribution centred on a known consensus $\boldsymbol{\rho}^0 = (1, ..., n)$, with different combinations of $\alpha^0 \in \{3, 5, 10\}$, $N \in \{500, 5000\}$ and $n \in \{50, 100, 1000\}$. For each setting of $\{\alpha^0, N, n\}$, 20 datasets are generated. For each dataset, we run the Bayesian Mallows MCMC and the Pseudo-Mallows, for a grid of different numbers of iterations, translating to different computation times. For all Bayesian Mallows MCMC runs reported in this paper we have used the BayesMallows package (Sørensen et al., 2020). After running the algorithms, we obtain a point estimate for the consensus parameter $\boldsymbol{\rho}$ by calculating the CP consensus (Sørensen et al., 2020) based on the samples obtained by the algorithm. Then we record the footrule distance between the CP consensus and the true $\boldsymbol{\rho}^0$ to assess the estimation accuracy of both algorithms for a given computing time.

The results for two of the scenarios are displayed in Figure 1, showing the footrule distance between the CP-consensus estimate and the true consensus $\boldsymbol{\rho}^0$ as the measure of accuracy on the $y$-axis, and computation time along the $x$-axis. The results for the other scenarios can be seen in Figure 11 in Appendix E, along with an additional simulation study. Consistently through the scenarios with $n \in \{50, 100\}$, the Bayesian Mallows MCMC's estimation accuracy initially improves as the algorithm runs for a longer period of time, until it stabilises after the algorithm has converged to the target distribution. For the more high-dimensional situation with $n = 1000$ (Figure 1, bottom panel), however, the MCMC does not reach convergence in the run time shown. The Pseudo-Mallows does not have a burn-in period, and accurate estimation can be achieved in a short amount of time. Not surprising, as the data comes from the Mallows distribution, and the Pseudo-Mallows is an approximation of the Bayesian Mallows MCMC, when convergence is reached by the Bayesian Mallows MCMC, its estimation of $\boldsymbol{\rho}$ is slightly more accurate compared to using the Pseudo-Mallows.

The efficiency of the Pseudo-Mallows is not due to the computation time per iteration, but rather to not needing a burn-in period, which instead forces the Bayesian Mallows MCMC to iterate much longer to reach convergence. Moreover, even after convergence is reached, many more MCMC samples than the desired sample dimension are needed due to the high autocorrelations between consecutive samples, and to the low acceptance rate. The Pseudo-Mallows samples are instead drawn independently, so that the desired sample dimension corresponds to what one actually needs to sample. As a result, much fewer samples are needed. In summary, due to these reasons, the Pseudo-Mallows algorithm is much faster compared to the Bayesian Mallows MCMC. Compared to the Bayesian Mallows MCMC, the Pseudo-Mallows can provide comparable estimation accuracy of the group consensus parameter in a much shorter

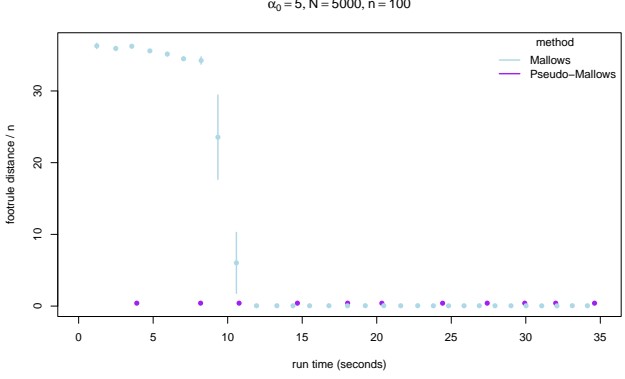

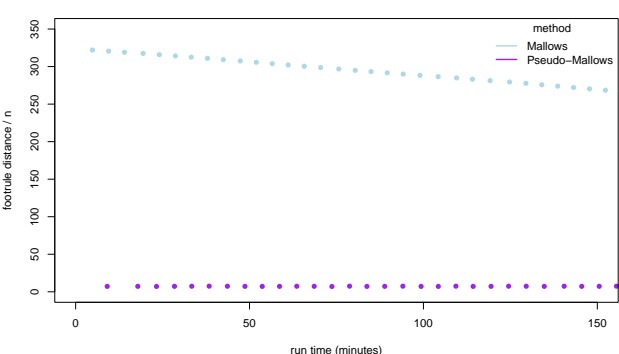

*Figure 1.* $y$-axis: mean posterior footrule distance to the true consensus $\boldsymbol{\rho}^0$ ($\pm$ posterior std dev); $x$-axis: run time, for two of the scenarios of the simulation study of Section 4.1.

period of time. This is especially beneficial when computation time is a priority.

## 4.2. Personalized recommendations from clicking data - a case study

In this section, we make personalized recommendations with the Pseudo-Mallows and the Mallows MCMC in a real life clicking dataset from the Norwegian Broadcasting Corporation (NRK), and compare recommendation performances.

The dataset contains $N = 7872$ anonymous users and $n = 200$ different movies, TV programs and news programs. Different episodes of the same TV series are considered as one item, and multiple clicks on an item by a user are only counted once. Each user has a minimum of 13 clicks. To construct a training dataset, we randomly remove 10 clicks per user, and these 10 clicks per user are later used as the ground truth test set for recommendation accuracy checking.

For this type of data, it is a natural assumption that there exist many user groups with different consensus rankings. Therefore, before applying the recommendation methods, we partition the dataset into $K = 17$ clusters using the

cluster assignment in line with the work of Liu et al. (2019b), with cluster sizes ranging from 35 users to 2801 users.

For each cluster, we then obtain an estimate of the scale parameter $\alpha$ following an extension for clicking data of the method for full ranking data presented in 2.4, which is sketched in Appendix D.3. For cluster 1, each user has in the training data clicked on an average of 8 items. We then generate full ranking datasets of $N = 500$ users from a Mallows distribution with $n = 200$ items, for a grid of known $\alpha^0$ values. These datasets are then binarized to get clicking data by drawing the number of clicks $c_j$ for user $j$ from a truncated Poisson distribution with parameter $\lambda = 8$, truncated to a minimum of 3. Thereafter, the top ranked items $c_j$ are considered clicked for user $j$, while the rest of the items considered unclicked. We then calculate the average pairwise individual similarity by (23) for each simulated dataset. Figure 2 shows a plot of the values of $\alpha^0$ and its corresponding pairwise user similarity. It can be observed that there appears to be a close to linear one-to-one relationship between $\alpha^0$ and the average pairwise user cosine similarity. The pairwise user similarity thus seems to be a good statistics to estimate $\alpha$. Lastly, we compute the pairwise user similarity in NRK cluster 1, and the $\alpha^0$ value of the simulated dataset with the closest pairwise user cosine similarity is chosen. We repeat this procedure for every cluster to obtain an estimate of $\alpha$ for all clusters.

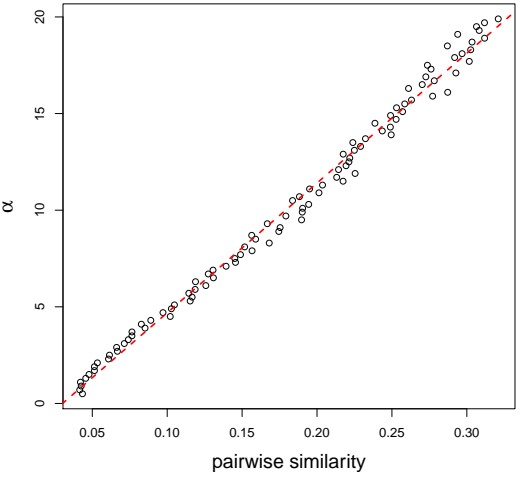

*Figure 2.* $\alpha$ tuning for cluster 1 of the NRK dataset. $x$-axis: average pairwise user cosine similarities, $y$-axis: corresponding $\alpha^0$ values from the full ranking dataset.

### 4.2.1. COMPARISON OF COMPUTATION SPEED AND RECOMMENDATION ACCURACY

After partitioning the dataset into 17 clusters, and estimating the $\alpha$ for each cluster, we run both the Mallows MCMC and the Pseudo-Mallows algorithm for each cluster. The algorithms are run for 10 minutes. We then make $K = 10$ recommendations per user with the samples obtained by both Mallows MCMC and Pseudo-Mallows. We compare the 10

recommendations per user made by the two algorithms to the true 10 clicks in the test set, and the intersection of the two sets gives the proportion of correct recommendations.

Recommendation accuracy of the two methods is shown in Table 1. It can be observed that, given the computing time limitation, the Pseudo-Mallows's recommendation accuracy is nearly double that of the Mallows MCMC. This is consistent with the simulation studies, showing that the Mallows MCMC suffers from the burn-in period as well as from highly autocorrelated samples, so that huge numbers of iterations are needed for the algorithm to reach full convergence. In the Pseudo-Mallows, instead, all samples are drawn independently, and accurate recommendations can be obtained with very few samples in much shorter time.

*Table 1.* Recommendation accuracies for the NRK data.

| Method | Mallows MCMC | Pseudo-Mallows |
|--------|--------------|----------------|
| Accuracy | 13.9% | 22.6% |

Not only can the Pseudo-Mallows make accurate recommendations rapidly, but its uncertainty estimation for each recommendation is also well calibrated. Each top-k recommendation has a posterior probability associated with it, as described in (14). We can bin all the top-$k$ recommendations according to these posterior probabilities, and calculate the recommendation accuracy (estimated with the proportion of correct recommendations) in each bin for both the Pseudo-Mallows and the Mallows MCMC. Figure 3 clearly shows that the Mallows MCMC systematically overestimates the posterior probabilities of the recommendations (red triangles). This is another sign that the Mallows MCMC algorithm has yet to reach convergence. On the other hand, the posterior probabilities estimated for each top-k recommendation ($x$-axis) by the Pseudo-Mallows largely reflect the actual recommendation accuracy, as a clear diagonal trend can be observed for the blue dots in Figure 3. The accurate uncertainty calibration also shows the effectiveness of the $\alpha$ tuning method proposed in Appendix D.3.

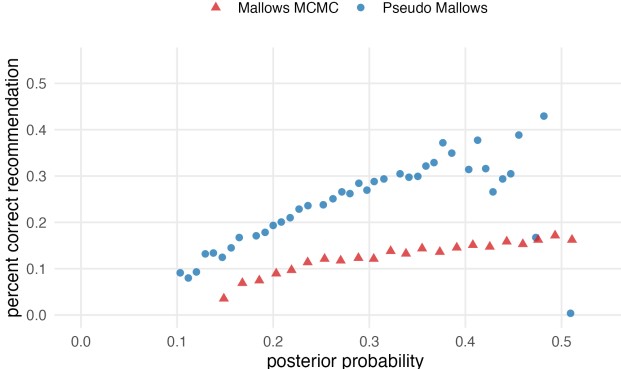

*Figure 3.* Comparison of top-k recommendation's uncertainty estimation for the NRK dataset for Mallows MCMC and Pseudu-Mallows. $x$-axis: posterior probability of top-k recommendations, $y$-axis: recommendation accuracy

## 5. Summary and discussion

In this work, we introduce the Pseudo-Mallows distribution to achieve a fast approximation of the Bayesian Mallows posterior distribution. The Pseudo-Mallows distribution is an ordered product of $n$ univariate distributions, where each variable is conditioned on all preceding ones. This ordering is the variational parameter which must be chosen optimally to obtain the best approximation. This is found by minimizing the marginalized version of the KL-divergence between the pseudo-Mallows distribution of all unknown parameters and the posterior Mallows distribution of such parameters given data. When the data are complete rankings of the $n$ items by each user, and the objective is inference on the consensus $\rho$, we argue that the optimal variational order is the $\mathcal{V}$-set based on the group consensus. We also show how the $\mathcal{V}$-set can be inferred from the data. We extend the Pseudo-Mallows method to learn the group consensus and all the individual rankings from clicking data, which is the most common situation for recommender systems.

Through a simulation study, we demonstrate that the algorithm based on the Pseudo-Mallows distribution can efficiently learn the group consensus parameter with few samples and short computation time. We also show for a real dataset that the Pseudo-Mallows for clicking data is indeed a faster alternative to Bayesian Mallows MCMC to learn individual rankings and hence make personalised recommendations.

There are several areas for further improvement. In applications such as personalized recommendations and rank aggregation, it is often a subset of items that is of interest, for example, the top-$k$ ranked items. In our current approach, we estimate the rankings of all items in each iteration, instead of focusing only on a subset of relevant items. In our Pseudo-Mallows algorithm, the ranking for each item is sampled sequentially, and it is possible to sample only a top part of the items, similarly to what has been proposed for the Bayesian Mallows MCMC in Eliseussen et al. (2021). Clearly, this would lead to further reducing the overall computation time. Also, the scale parameter $\alpha$ is tuned by cosine similarity in our work. Instead, we could optimise it as a hyperparameter, for example by Bayesian optimization (Alvi et al., 2019) or by commonly used variational inference optimization schemes such as CAVI or SVI.

In conclusion, the Pseudo-Mallows method with our proposal of the optimal variational parameters leads to a successful approximation of the Bayesian Mallows posterior, is computationally convenient and leads to faster inference. This opens for a broader use of the Bayesian Mallows-based recommender for personalized preference learning.

## Acknowledgements

This work was supported by the Research Council of Norway through its Centre of Excellence Integreat - The Norwegian Centre for knowledge-driven machine learning, project number 332645. We sincerely thank the four anonymous reviewers for their thoughtful suggestions and comments, which helped us significantly improve the clarity and quality of the manuscript.

## Impact Statement

By significantly lowering the computational burden of the Mallows recommender system, this work enables more efficient and broader use of the Mallows-based probabilistic recommender. Combined with careful evaluation of fairness and privacy issues, which are easier in white-box model-based methods, the proposed approach can accelerate access to personalization while reducing real-world negative effects (spam), environmental burden (CPU time) and monetary costs.

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

# Appendix

## A. The Mallows ranking model

The Mallows is an exponential family distribution, which assumes that the latent individual permutations of the items, which we call rankings, are centred around a group consensus $\boldsymbol{\rho} \in \mathcal{P}_n$. A distance between permutations measures the closeness between each user's ranking and $\boldsymbol{\rho}$. The dispersion of the users' rankings from the consensus, their individual inclination, is controlled by a scale parameter $\alpha$. Several right invariant distances are considered in Vitelli et al. (2018), among which the $l_1$-norm, called footrule, and the $l_2$-norm, called the Spearman distance. For a group of homogeneous users with rankings $\{\mathbf{R}^j\}_{j=1}^N$ that share their preferences to some degree, the Mallows model assumes that their rankings are distributed around a consensus parameter $\boldsymbol{\rho}$, with a scale parameter $\alpha$. Assuming that all users' rankings are mutually independent given $\boldsymbol{\rho}$ and $\alpha$, the Mallows likelihood is given by

$$P(\mathbf{R}^1, ..., \mathbf{R}^N | \alpha, \boldsymbol{\rho}) = \prod_{j=1}^N \frac{\exp\{-\frac{\alpha}{n} \boldsymbol{d}(\mathbf{R}^j, \boldsymbol{\rho})\}}{Z_n(\alpha)} = \frac{\exp\{-\frac{\alpha}{n} \sum_{j=1}^N \boldsymbol{d}(\mathbf{R}^j, \boldsymbol{\rho})\}}{Z_n(\alpha)^N}, \tag{15}$$

where $\boldsymbol{d}(,)$ is a right-invariant distance (Diaconis, 1988) between two rankings, and $Z_n(\alpha)$ is the normalizing constant, which does not depend on $\boldsymbol{\rho}$ when the distance is right-invariant. We use the right-invariant footrule distance, defined as $\boldsymbol{d}(\mathbf{R}^j, \boldsymbol{\rho}) = \sum_{i=1}^n d(R_i^j, \rho_i) = \sum_{i=1}^n |R_i^j - \rho_i|$. The parameter of interest for inference is the consensus $\boldsymbol{\rho}$. Given $\alpha$, the Mallows posterior distribution of the consensus parameter is

$$P(\boldsymbol{\rho} | \alpha, \mathbf{R}^1, ..., \mathbf{R}^N) = \frac{P(\mathbf{R}^1, ..., \mathbf{R}^N | \alpha, \boldsymbol{\rho})\pi(\boldsymbol{\rho})}{\sum_{\boldsymbol{r} \in \mathcal{P}_n} P(\mathbf{R}^1, ..., \mathbf{R}^N | \alpha, \boldsymbol{r})\pi(\boldsymbol{r})},$$

where $\mathcal{P}_n$ is the space of permutation of $n$ items, and $\pi(\boldsymbol{\rho})$ is a prior distribution for $\boldsymbol{\rho}$. For simplicity, we assume a non-informative uniform prior for $\pi(\boldsymbol{\rho})$, i.e., $\pi(\boldsymbol{\rho}) = \frac{1}{n!}$. Then,

$$P(\boldsymbol{\rho} | \alpha, \mathbf{R}^1, \ldots, \mathbf{R}^N) = \frac{\exp\left\{-\frac{\alpha}{n} \sum_{j=1}^N d(\mathbf{R}^j, \boldsymbol{\rho})\right\}}{Z_n(\alpha, \mathbf{R}^1, \ldots, \mathbf{R}^N)}, \tag{16}$$

where

$$Z_n(\alpha, \mathbf{R}^1, \ldots, \mathbf{R}^N) = \sum_{\mathbf{r} \in \mathcal{P}_n} \exp\left\{-\frac{\alpha}{n} \sum_{j=1}^N d(\mathbf{R}^j, \mathbf{r})\right\}.$$

MCMC algorithms can be used to make inference on $\boldsymbol{\rho}$ as in Vitelli et al. (2018). Summary statistics such as the Maximum A Posteriori (MAP) estimators can be computed based on the MCMC samples. The MCMC scheme proposed by Vitelli et al. (2018) can perform well when there are not many items and users. However, the MCMC suffers from slow convergence. In order to achieve good mixing and precise inferential results, the MCMC chain needs to be run for very long. Because the posterior distribution has multiple peaks, the MCMC is "sticky" (Gilks & Roberts, 1996), i.e., the chain gets easily stuck at a local optima. The "stickiness" problem is pronounced because the $\boldsymbol{\rho}$ parameter is discrete.

## B. Theoretical and empirical justification of the Optimal Pseudo-Mallows Conjecture

### B.1. Intuition of the Optimal Pseudo-Mallows Conjecture

The reason why the "V"- rankings are the best choice for the way to factorize the Pseudo-Mallows distribution is because this makes the Pseudo-Mallows distribution the most similar to the Mallows posterior distribution, hence maximize the

ELBO, when the data comes from a Mallows distribution. For two distributions to be identical, it is necessary that their modes coincide. Assuming this data generating process, i.e., $\mathbf{R}^j \sim$ Mallows $(\boldsymbol{\rho}^0, \alpha)$, $j = 1, ..., N$, the mode of the Mallows posterior is $\boldsymbol{\rho}^0$. In order to make the Pseudo-Mallows similar to the Mallows posterior, we want the mode of the Pseudo-Mallows distribution to be as close to $\boldsymbol{\rho}^0$ as possible.

Recall that the $k$-th term in the Pseudo-Mallows distribution, i.e.,

$$q(\rho_{o_k}|\alpha, o_k, \rho_{o_1}, ..., \rho_{o_{k-1}}, R^1_{o_k}, ..., R^N_{o_k})$$

has the form (3). The mode of this is the value $l^*$ that minimizes the distances in the numerator, i.e.,

$$l^* = \underset{l \in \{1,..,n\} \setminus \{\rho_{o_1}, ..., \rho_{o_{k-1}}\}}{\operatorname{argmin}} \sum_{j=1}^{N} d(R^j_{o_k}, l)\}. \tag{17}$$

We simplify the above minimization by considering the expected distance with respect to the Mallows distribution, i.e., $l^* = \operatorname{argmin} \mathbb{E}[\sum_{j=1}^{N} d(R^j_{o_k}, l)\}]$. For the $k$-th term in (3), this expected distance is minimized by the median of $P_{Mallows}(R_{o_k}|\alpha, \boldsymbol{\rho}^0)$, i.e., the marginal Mallows distribution of $R_{o_k}$, or more formally,

$$l^* = \underset{l \in \{1,..,n\} \setminus \{\rho_{o_1}, ..., \rho_{o_{k-1}}\}}{\operatorname{argmin}} \mathbb{E}[\sum_{j=1}^{N} d(R^j_{o_k}, l)\}] = \operatorname{median}\left(P_{Mallows}(R_{o_k}|\alpha, \boldsymbol{\rho}^0)\mathbb{1}_{R_{o_k} \neq \rho_{o_1}, ..., \rho_{o_{k-1}}}\right), \tag{18}$$

as proven in Appendix G.1, and hence we want $l^*$ to be similar to $\rho^0_{o_k}$.

Let us first consider the first term $k = 1$ in (3). Recall that item $o^0_k$ is the item with rank $k$ in $\boldsymbol{\rho}^0$. For the item with middle rank in $\boldsymbol{\rho}^0$, i.e., item $o^0_m$, its ranking $R_{o^0_m}$ is distributed symmetrically about $m$, therefore, the median of $R_{o^0_m}$ is equal to its expected value, which is exactly $m$. Detailed proof of this symmetry and expected value is shown in Appendix G.2. Based on this property, by choosing item $o^0_m$ as the first factor in (3), we ensure that the mode of $\rho_{o^0_m}$ coincides with the wished target $m$.

However, this special property of the median holds only for the middle item $o^0_m$. Let us take a closer look at the properties of the median of each $R_{o^0_k}$, $k = 1, ..., n$. In Figure 4 we study the marginal median of $R_{o^0_k}$ as a function of $\alpha$, and compare them for all $k$. We approximate the medians by drawing 50000 independent samples from the Mallows distribution centred at $\boldsymbol{\rho}^0 = (1, 2, ..., 9)$ (hence in this case, $o^0_k = k$ for $k = 1, ..., n$ ), on a grid of $\alpha^0$ values. We plot the empirical marginal median of $R_k$ on the y-axis, against the corresponding $\alpha^0$ value. It can be observed the median for $R_{o_k}, k \neq 5$, is a function of $\alpha^0$, and converges to $k$ for $\alpha^0$ increasing, and towards the middle rank 5 as $\alpha^0$ decreases to 0. In other words, for all $k \neq 5$, for low enough values of $\alpha^0$, the median of $R_k$ is away from the corresponding rank in $\boldsymbol{\rho}^0$, i.e., $k$ in this case, and deviates further and further from $k$ (towards the middle value 5) as $\alpha^0$ decreases.

We also observe that the deviation of the median from its corresponding rank in $\boldsymbol{\rho}^0$ is the most pronounced for the items with top- and bottom-ranks in $\boldsymbol{\rho}^0$, i.e., item 1 and 9 in this example. In the case of very small $\alpha^0$, the medians of item 1 and 9 can deviate 4 positions from $\rho_1$ and $\rho_9$ respectively, whereas item 4 and 6's medians deviate only by 1 position.

Based on these properties, we conjecture that the optimal sequence for the terms in (3) is by starting with the middle item $o^0_m$, and then outwards i.e., first the middle item $o^0_m$, then the two second middle-ranked items (items $o^0_{m-1}$ and $o^0_{m+1}$) and so on until lastly the two edge items. By doing so, we first sample for the item whose marginal median (and hence mode) is the closest to the targeted value in $\boldsymbol{\rho}^0$. After a sample is drawn, the value that it takes on will be excluded for the subsequent items. As shown in (18), due to $l^*$ being the median of $R_{o_k}|R_{o_k} \neq \rho_{o_1}, ..., \rho_{o_{k-1}})$, this forces the subsequent items to be centered closer to their corresponded rank in $\boldsymbol{\rho}^0$. This explains why a $\delta$-distribution with its mass concentrated on any ordering that belongs to the $\mathcal{V}_{\boldsymbol{\rho}^0}$-set helps the Pseudo-Mallows distribution to best approximate the Mallows posterior.

### B.2. Empirical study of the Optimal Pseudo-Mallows Conjecture

To support the Optimal Pseudo-Mallows conjecture, we conduct an empirical study where the optimization problem (6) is solved first by enumeration for a small number $n$ of items, and then by means of a local search algorithm for larger $n$. In these cases, we show that a ranking $\boldsymbol{v} = (v_1, ..., v_n) \in \mathcal{V}_{\boldsymbol{\rho}^0}$ minimizes the KL-divergence between the Pseudo-Mallows

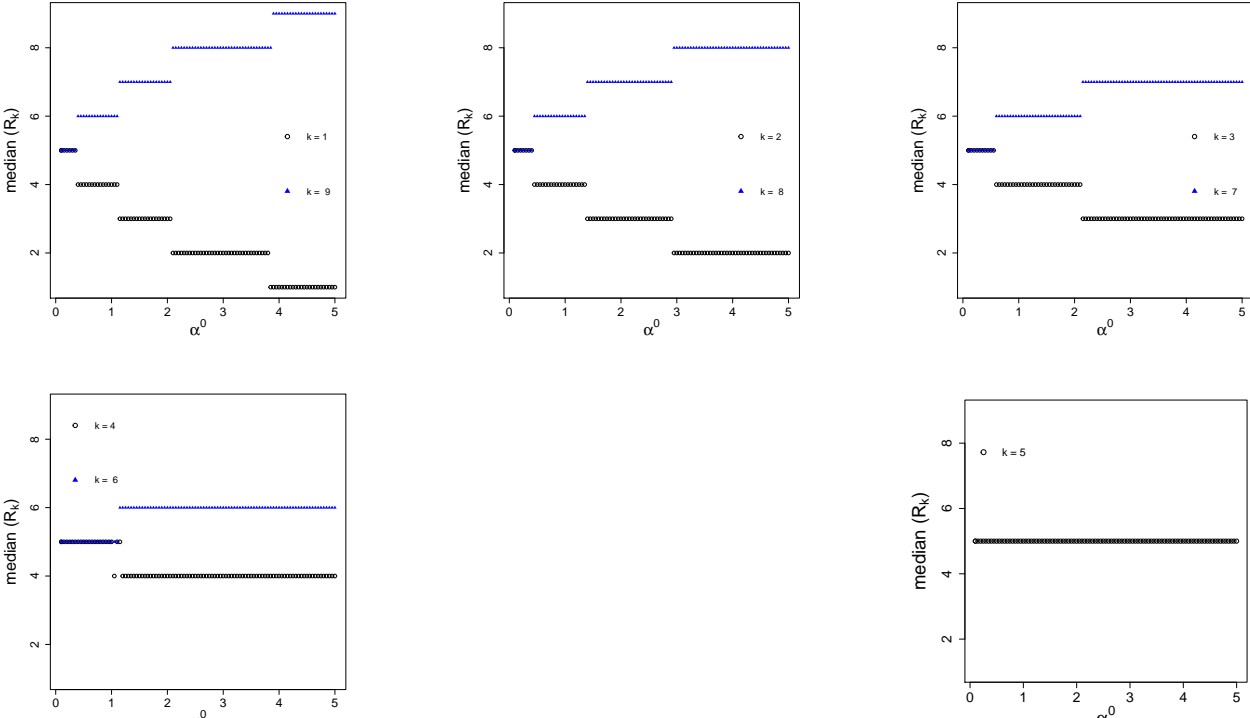

*Figure 4.* Marginal median of $R_{o_k^0}$ where $\mathbf{R}$ is drawn from a Mallows model centred at $(1, 2, ..., n)$. A similar figure can be obtained for any other values of $n$.

distribution and the Mallows posterior. Computing the KL-divergence (6) between two distributions over $\mathcal{P}_n$ is difficult for a large $n$ because of the exponential dimensions of the set of permutations. In this paper we use instead the sum of one-dimensional KL-divergences between the Mallows posterior's and the Pseudo-Mallows marginal distributions of each item. More formally, we work with the sum of marginal KL-divergences defined as

$$\sum_{i=1}^{n} \text{KL}\left(q(\rho_i|\alpha, \mathbf{R}^1, ..., \mathbf{R}^N, \boldsymbol{v})||P(\rho_i|\alpha, \mathbf{R}^1, ..., \mathbf{R}^N)\right), \tag{19}$$

as measure of discrepancy.

**Enumeration for $n \leq 9$** When the number of items $n$ is small, it is possible to enumerate all possible rankings $\boldsymbol{v} \in \mathcal{P}_n$, and compute the KL-divergence between the Pseudo-Mallows distribution and the Mallows posterior. We first generate full ranking datasets with $n$ items and $N$ users by drawing $N$ independent rankings from the Mallows distribution with $\alpha^0$ and $\boldsymbol{\rho}^0$ using the BayesMallows R package (Sørensen et al., 2020). For convenience, we fix $\boldsymbol{\rho}^0 = \{1, 2, ..., n\}$. Next, we run the MCMC converging to the Mallows posterior to obtain samples. Then for every ranking $\boldsymbol{v} \in \mathcal{P}_n$ we draw 200 independent samples from the Pseudo-Mallows distribution given $\boldsymbol{v}$ using Algorithm 1 in Appendix F. We then calculate the marginal KL-divergence (19) between the Pseudo-Mallows distribution and the Mallows posterior. As the number of items $n$ is small, we carefully choose the $\{\alpha^0, N, n\}$ combination to avoid that the Mallows posterior distribution is too concentrated or too uniform. For each such combination of $\{\alpha^0, N, n\}$, we generate 20 datasets, and for each dataset, we record the ranking(s) $\boldsymbol{v}$ that lead to the lowest marginal KL-divergence during each run. The distribution of the optimal rankings from the 20 runs are shown in a heat plot in Figure 5. Item 1, ..., $n$ are shown on the x-axis, and their probabilities to be ranked as $\{1, (2i \text{ or } 2i+1), i = 1, ..., (n-1)/2\}$ are shown on the y-axis. As demonstrated in Figure 5, rankings $\boldsymbol{v}$ such that the middle item is top-ranked, the two items next to the middle are ranked second and third, and so on, until lastly the items "on the edge" are bottom-ranked, appear to be the rankings that minimize the marginal KL-divergence between the Pseudo-Mallows and the Mallows posterior, as conjectured. The heat plots show that these rankings present a "V" shape, which is the reason of the name "V"-rankings.

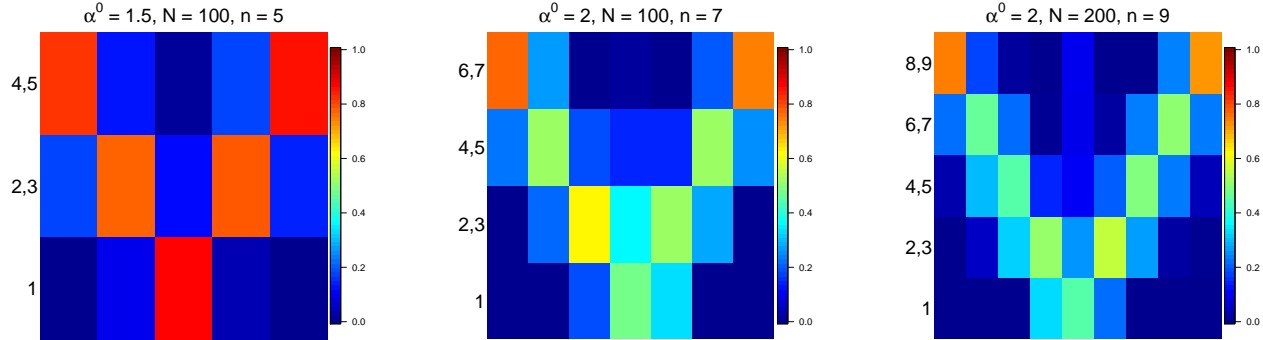

*Figure 5.* Heat plots of $\boldsymbol{v}$ rankings that result in the lowest KL-divergence. The bottom row represents all items' probability to be ranked as 1, the second row from the bottom represents all items' probability to be ranked as either 2 or 3, etc.

**Iterated search for moderate number of items.** For the case in which $n$ is such that enumeration of all possible $\boldsymbol{v}$ rankings is infeasible, to minimize the marginal KL we propose a heuristic approach based on an iterative optimization algorithm. When we apply this algorithm to some examples we find that the "V"-rankings consistently give rise to minimum KL-divergence.

The idea of our iterative algorithm is to compute a sequence of permutations $\boldsymbol{v}^1, \boldsymbol{v}^2, \ldots$, trying to improve the marginal KL at each iteration. To this end, we define a sequence of objective functions $w^1, w^2, \cdots : N \to \mathbb{R}$ such that, at iteration $l = 1, 2, \ldots$, (i) $\boldsymbol{v}^l$ is the optimal permutation with respect to $w^l$ and (ii) $\boldsymbol{v}^l$ can be computed efficiently because we are using the marginal KL. The function $w^l$ is defined by perturbing the original objective (19) according to the previous solution $\boldsymbol{v}^{l-1}$ (we assume an initial solution $\boldsymbol{v}^0$ is given).

We first describe the form of the functions $w^1, w^2, \ldots$ and how we can effectively find the optimal permutation $\boldsymbol{v}^l = \{v_1^l, \ldots, v_n^l\}$ at each iteration. We then specify how $w^l$ is built at iteration $l$ from $\boldsymbol{v}^{l-1}$.

For each item $j \in N$ and each iteration $l$, we let $w_{jr}^l$ represent some approximation of the marginal cost of assigning rank $r$ to $j$ (i.e. $v_r^l = j$). Then we define the cost of the permutation $\{v_1, \ldots, v_n\}$ simply as

$$w^l(\{v_1, \ldots, v_n\}) = \sum_{r \in N} w_{v_r, r}^l \qquad (20)$$

Computing the permutation which minimizes (20) is equivalent to finding a minimum weighted perfect matching in a suitable (complete, balanced) bipartite graph (Cook & Rohe, 1999), and it can be performed in polynomial time (in $|N|$).

We are now left with describing how $w_{j,r}^l$ is defined at each iteration $l = 1, 2, \ldots$, for any $j, r \in N$. As said, this quantity represents an approximation of the increase of cost (w.r.t. the previous permutation $\boldsymbol{v}^{l-1}$) when item $j$ is assigned rank $r$ in $\boldsymbol{v}^l$. Finding the exact value for $w_{j,r}^l$ is per se as difficult as the original optimization problem, as it amounts to find the optimal permutation when $j$ has fixed rank $r$. So, we content ourselves with an approximate solution of this hard problem, following an idea developed in (Vitelli et al., 2018). Namely, we heuristically build a permutation $\mu^{l-1}$ from $\boldsymbol{v}^{l-1}$ by assigning to $j$ rank $r$ and by "locally" re-adjusting the ranking of a subset of items. Then we let $w_{j,r}^l$ equal the difference in cost (computed as (19)) between $\mu^{l-1}$ from $\boldsymbol{v}^{l-1}$.

Namely, we let:

$$w_{jr}^l = \text{KL}\Big(q\big(\,\cdot\,|L\&S(\{v_1^{l-1}, ..., v_n^{l-1}\}, j, r))||p(\cdot)\big) - \text{KL}\Big(q\big(\,\cdot\,|v_1^{l-1}, ..., v_n^{l-1}))||p(\cdot)\big),$$

where $L\&S(\{v_1^{l-1}, ..., v_n^{l-1}\}, j, r)$ is a variation of the "Leap & Shift" proposal introduced in Vitelli et al. (2018), described as follows: for a given ranking $\{v_1^{l-1}, ..., v_n^{l-1}\}$, the item $j$ to be perturbed with current ranking $q = v_j^{l-1}$ and destination

ranking $r = v_j^l$, we let $L\&S(\{v_1^l, ..., v_n^l\}, j, r)$ be the following ranking for $m = 1, ..., n$

$$v_m^l = \begin{cases} j, & \text{if } m = r \\ v_m^{l-1} - 1, & \text{if } q < v_m^{l-1} \le r \text{ and } q < r \\ v_m^{l-1} + 1, & \text{if } r \le v_m^{l-1} < q \text{ and } q > r \\ v_m^{l-1}, & \text{otherwise} \end{cases} .$$

At each iteration $l$, once all weights $w^l$ are computed as above, we use the Hungarian Method (Kuhn, 1955) to compute the minimum weight perfect matching, which provides us the new permutation $\boldsymbol{v}^{l+1}$. The search algorithm will run iteratively until a stop criterion, such as a maximum number of iteration, is met. In practice, we recommend setting the maximum number of iterations of the algorithm to a large enough value, depending on $n$. Clearly, the computational effort of the algorithm quickly increases with the number of items $n$. The iterated search algorithm is more formally described in Algorithm 3 in Appendix F.

In order to simplify the verification of the "V"-ranking's optimality, we initialize the algorithm by generating a "V"-ranking, and then conduct $n$ swaps steps with its neighbor so that the starting point is not a "V"-ranking, but not too far away from the "V"-ranking.

Similarly to the full enumeration study, we generate 20 full ranking datasets for each combination of $\{\alpha^0, N, n\}$, and run Algorithm 3 in Appendix F on each dataset. The maximum number of iterations of the algorithm was set to 100 when $n$ was small, to 50 otherwise. In Figure 6, we have selected some runs to demonstrate the characteristics of the algorithm. The blue dots represent the marginal KL-divergence, while the red triangles represent the footrule distance between the current ranking and a "V"-ranking. As "V"-rankings are not unique, we use the distance between the current ranking and the "V"-ranking closest to it. From Figure 6, it can be observed that the iterative search algorithm has a tendency to "re-start" occasionally: i.e., after finding a permutation that leads to a locally-minimum marginal KL, it can move away from it. However, consistently with the enumeration experiment, when the algorithm discovers a "V"-ranking, the corresponding KL-divergence is typically very low.

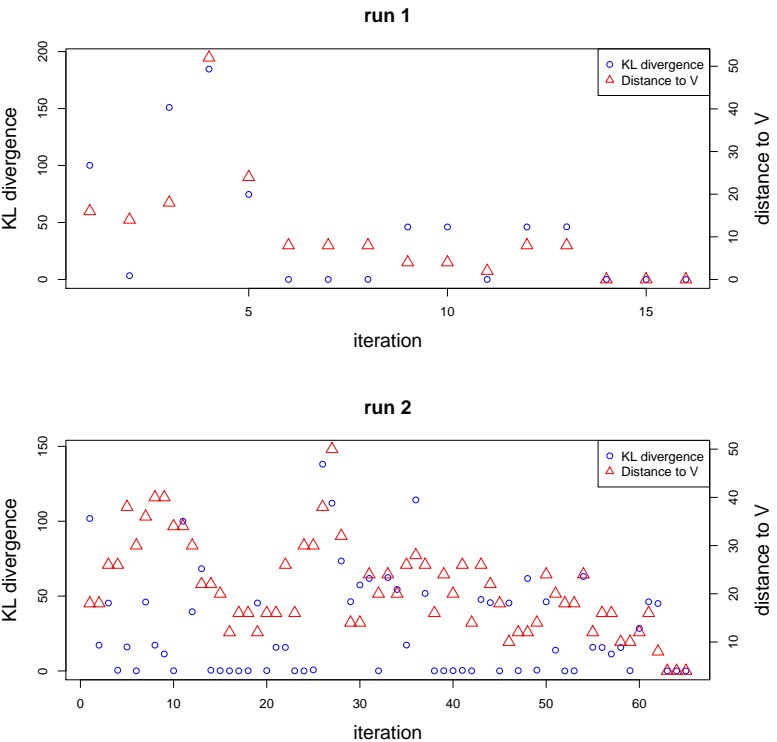

*Figure 6.* Selected runs of the iterative search algorithm, $n$= 15, $N = 500$, $\alpha^0 = 2.5$

We record the $\boldsymbol{v}$ ranking of the last iteration of the algorithm at each run, and aggregate these rankings in a heat plot, as shown in Figure 7. It can be clearly observed that the iterative search algorithm typically stops exactly at a "V"-ranking, which is consistent with our conjecture. Sometimes however, the algorithm did not stop exactly at a "V"-ranking, but with a short footrule distance therefrom, despite a very similar marginal KL-divergence. This is due to the approximation of the marginal KL-divergence and the finite sample situation.

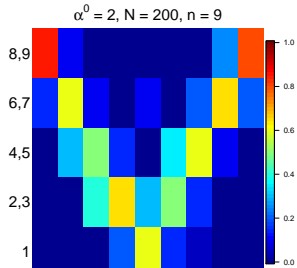 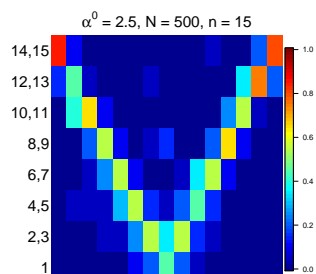 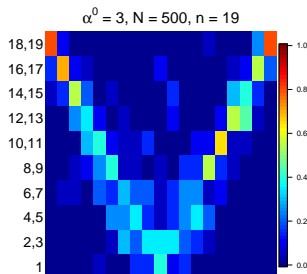

*Figure 7.* Iterative search algorithm's last iteration, aggregated over 20 runs

## C. Inferring the $\mathcal{V}$-set from data: performance of the estimation procedure for finite $N$

The performance of the inferential procedure for estimating the $\mathcal{V}$-set from data is illustrated in Figure 8. For both the cases of $\alpha^0 = 1$ and $\alpha^0 = 2$, introducing a perturbation with $\sigma > 0$, the marginal KL-divergence between the Pseudo-Mallows and the Mallows posterior is reduced. It can also be observed that as $\alpha^0$ increases, $N$ increases and $n$ decreases, the $\sigma$ value that resulted in a reduction of the marginal KL-divergence decreases. In some situations, the optimal value for $\sigma$ can be 0.

Figure 9 illustrates how the optimal $\sigma$ changes with respect to $\alpha^0$, $N$ and $n$. In each panel, for the chosen $N$ and $n$, we generate datasets by sampling from the Mallows distribution, on a grid of different values of $\alpha^0$. Ten datasets are generated for each value of $\alpha^0$. For each of these datasets, a grid of $\sigma$ values is tested, and the $\sigma$ value that results in the smallest marginal KL-divergence is plotted on the y-axis- resulting in 10 values for each value of $\alpha$ on the x-axis. We see how the optimal $\sigma$ depends on $\alpha^0$, $N$ and $n$. In particular, $\sigma$ decreases towards 0 as $N$ increases and $\alpha^0$ increases, and as $n$ decreases. When $\alpha^0$ and $N$ are sufficiently large and/or $n$ is sufficiently small, the different choices of $\sigma$ do not affect the KL-divergence. We can in other words safely use $\sigma = 0$ in such situations.

## D. Additional details for Pseudo-Mallows for clicking data

**D.1. Factorization of $q^R(\cdot)$**

$$
\begin{aligned}
&q^R(\mathbf{R}^j_{\mathcal{A}^j}|\alpha, \boldsymbol{\rho}^{t,j}, \boldsymbol{v}^{t,j})\\
=&q^R(\mathbf{R}^j_{\mathcal{A}^j}|\alpha, \boldsymbol{\rho}^{t,j}, \boldsymbol{o}^{t,j})\\
=&q^R(R^j_{o^{t,j}_1}|\alpha, \rho^{t,j}_{o^{t,j}_1}, o^{t,j}_1) \cdot q^R(R^j_{o^{t,j}_2}|\alpha, \rho^{t,j}_{o^{t,j}_2}, o^{t,j}_2, R^j_{o^{t,j}_1}) \cdot ...\cdot\\
&q^R(R^j_{o^{t,j}_{c_j-1}}|\alpha, \rho^{t,j}_{o^{t,j}_{c_j-1}}, o^{t,j}_{c_j-1}, R^j_{o^{t,j}_1}, ..., R^j_{o^{t,j}_{c_j-2}}) \cdot q^R(R^j_{o^{t,j}_{c_j}}|\alpha, \rho^{t,j}_{o^{t,j}_{c_j}}, o^{t,j}_{c_j}, R^j_{o^{t,j}_1}, ..., R^j_{o^{t,j}_{c_j-1}});\\
&q^R(\mathbf{R}^j_{\mathcal{A}^{jc}} - c_j|\alpha, \boldsymbol{\rho}^{b,j}, \boldsymbol{v}^{b,j})\\
=&q^R(\mathbf{R}^j_{\mathcal{A}^{jc}} - c_j|\alpha, \boldsymbol{\rho}^{b,j}, \boldsymbol{o}^{b,j})\\
=&q^R(R^j_{o^{b,j}_1}|\alpha, \rho^{b,j}_{o^{b,j}_1}, o^{b,j}_1) \cdot q^R(R^j_{o^{b,j}_2} - c_j|\alpha, \rho^{b,j}_{o^{b,j}_2}, o^{b,j}_2, R^j_{o^{b,j}_1}) \cdot ...\cdot\\
&q^R(R^j_{o^{b,j}_{c^c_j-1}} - c_j|\alpha, \rho^{b,j}_{o^{b,j}_{c^c_j-1}}, o^{b,j}_{c^c_j-1}, R^j_{o^{b,j}_1}, ..., R^j_{o^{b,j}_{c^c_j-2}}) \cdot q^R(R^j_{o^{b,j}_{c^c_j}}|\alpha, \rho^{b,j}_{o^{b,j}_{c^c_j}}, o^{b,j}_{c^c_j}, R^j_{o^{b,j}_1}, ..., R^j_{o^{b,j}_{c^c_j-1}});
\end{aligned}
\tag{21}
$$

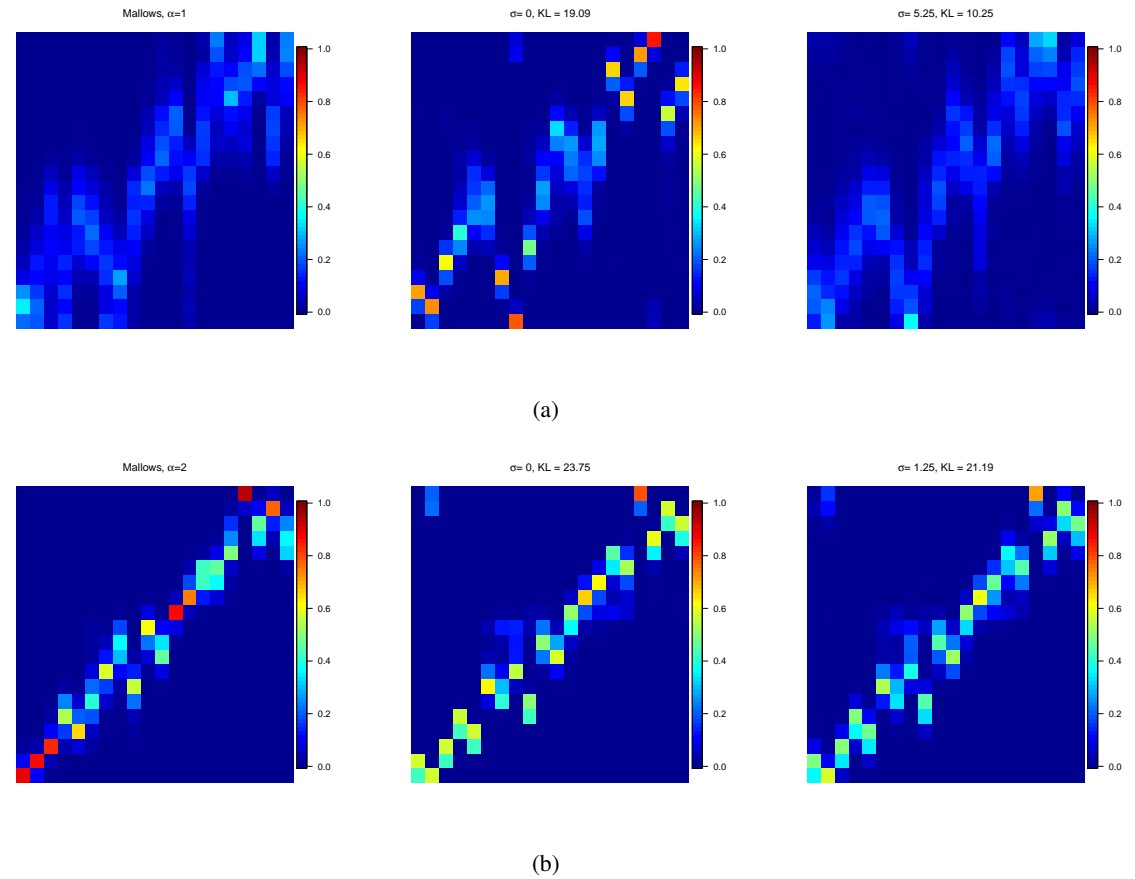

(a)

(b)

*Figure 8.* Left column: Mallows posterior; middle column: pseudo-Mallows with $\sigma = 0$; right column: pseudo-Mallows with $\sigma > 0$. $N = 100, n = 20$. (a) $\alpha^0 = 1$ and (b) $\alpha^0 = 2$.

where

$$q^R(R_{o_1^{t,j}}^j | \alpha, \rho_{o_1^{t,j}}^{t,j}, o_1^{t,j}) = \frac{\exp\{-\frac{\alpha}{n}d(R_{o_1^{t,j}}^j, \rho_{o_1^{t,j}}^{t,j})\}}{\sum\limits_{r \in \{1,..,c_j\}} \exp\{-\frac{\alpha}{n}d(r, \rho_{o_1^{t,j}}^{t,j})\}} \mathbb{1}_{R_{o_1^{t,j}}^j \in \{1,...,c_j\}}$$

$$q^R(R_{o_k^{t,j}}^j | \alpha, \rho_{o_k^{t,j}}^{t,j}, o_k^{t,j}, R_{o_1^{t,j}}^j, ..., R_{o_{k-1}^{t,j}}^j) = \frac{\exp\{-\frac{\alpha}{n}d(R_{o_k^{t,j}}^j, \rho_{o_k^{t,j}}^{t,j})\}}{\sum\limits_{r \in \{1,..,c_j\} \setminus \{R_{o_1^{t,j}}^j, ..., R_{o_{k-1}^{t,j}}^j\}} \exp\{-\frac{\alpha}{n}d(r, \rho_{o_k^{t,j}}^{t,j})\}}$$

$$\cdot \mathbb{1}_{R_{o_k^{t,j}}^j \in \{1,...,c_j\} \setminus \{R_{o_1^{t,j}}^j, ..., R_{o_{k-1}^{t,j}}^j\}}, \text{ for } k = 2, ..., c_j;$$

$$q^R(R_{o_1^{b,j}}^j - c_j | \alpha, \rho_{o_1^{b,j}}^{b,j}, o_1^{b,j}) = \frac{\exp\{-\frac{\alpha}{n}d(R_{o_1^{b,j}}^j, \rho_{o_1^{b,j}}^{b,j})\}}{\sum\limits_{r \in \{1,..,c_j^c\}} \exp\{-\frac{\alpha}{n}d(r, \rho_{o_1^{b,j}}^{b,j})\}} \mathbb{1}_{R_{o_1^{b,j}}^j \in \{1,...,c_j^c\}}$$

$$q^R(R_{o_k^{b,j}}^j - c_j | \alpha, \rho_{o_k^{b,j}}^{b,j}, o_k^{b,j}, R_{o_1^{b,j}}^j, ..., R_{o_{k-1}^{b,j}}^j) = \frac{\exp\{-\frac{\alpha}{n}d(R_{o_k^{b,j}}^j, \rho_{o_k^{b,j}}^{b,j})\}}{\sum\limits_{r \in \{1,..,c_j^c\} \setminus \{R_{o_1^{b,j}}^j, ..., R_{o_{k-1}^{b,j}}^j\}} \exp\{-\frac{\alpha}{n}d(r, \rho_{o_k^{b,j}}^{b,j})\}}$$

$$\cdot \mathbb{1}_{R_{o_k^{b,j}}^j \in \{1,...,c_j^c\} \setminus \{R_{o_1^{b,j}}^j, ..., R_{o_{k-1}^{b,j}}^j\}}, \text{ for } k = 2, ..., c_j^c;$$

(22)

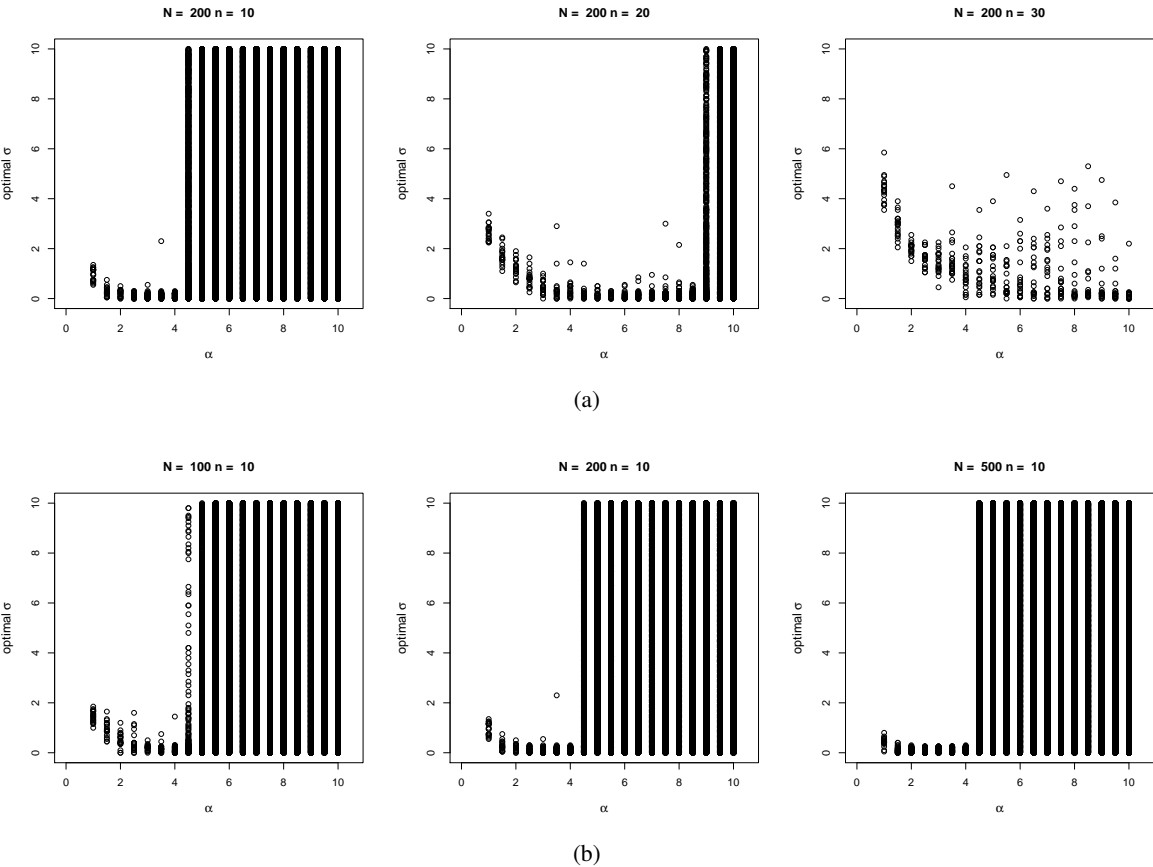

*Figure 9.* The optimal $\sigma$ value as a function of $\alpha$, for different combinations of values of $N$ and $n$.

## D.2. $g(\cdot)$ distributions for individual data augmentation

Indeed, the uniform distribution for $g(\cdot)$ gives the best approximation to the Mallows posterior distribution for user $j$'s full ranking $\mathbf{R}^j$, in terms of maximizing the ELBO, given that the rankings underlying the clicks come from a Mallows distribution. The intuitive motivation is similar to the reasoning why the $\mathcal{V}$-set construction gives rise to the best approximation for the consensus parameter $\rho$ when full individual rankings are available. However in this case, considering the clicked group, when inferring $\mathbf{R}^j$ using (13) the mode of the $k$-th term of (22) is $\rho_{o_k}$, for all $k$. For the unclicked group, the argument follows the same logic. Hence, no systematic rules regarding the sequence following which items should be sampled for (such as the $\mathcal{V}$-set construction) are needed, and the uniform distribution is the most effective choice. In fact, systematic rules would actually introduce systematic bias, as will be shown in the following experiment.

To conduct this experiment, we draw independent full rankings $\mathbf{R}^j$ from (12), and to illustrate the impact of different $g(\cdot)$ distributions in the augmentation step, we use different $g(\cdot)$ functions in the sampling, with known $\alpha^0$ and $\rho^0$. We also draw independent full ranking samples using the Mallows likelihood with the same parameters $\alpha^0$ and $\rho^0$, and monitor how the distribution of the full ranking samples resulting from different $g(\cdot)$ distributions differ from the distribution based on the Mallows samples.

For each $g(\cdot)$ distribution, we sample for $N = 1000$ full rankings of $n = 20$ items based on $\alpha^0 = 15$ and $\rho^0 = \{1, 2, ..., n\}$. The different $g(\cdot)$ distributions that we consider are:

- Uniform: for each $\mathbf{R}^j$ sample, we sample one random ranking uniformly from $\mathcal{P}_n$

- V: for each $\mathbf{R}^j$ sample, one V-ranking $v$ is sampled uniformly from $\mathcal{V}_{\rho^0}$

- top_bottom: $v = \rho^0$ for all $\mathbf{R}^j$ samples

Based on the $\mathbf{R}^j$ samples, we estimate the marginal distribution of each item $P(R_i^j|\alpha^0, \boldsymbol{\rho}^0)$, and we show the heat plots of these marginal distributions in Figure 10. The items are arranged according to their rankings in $\boldsymbol{\rho}^0$ on the x-axis.

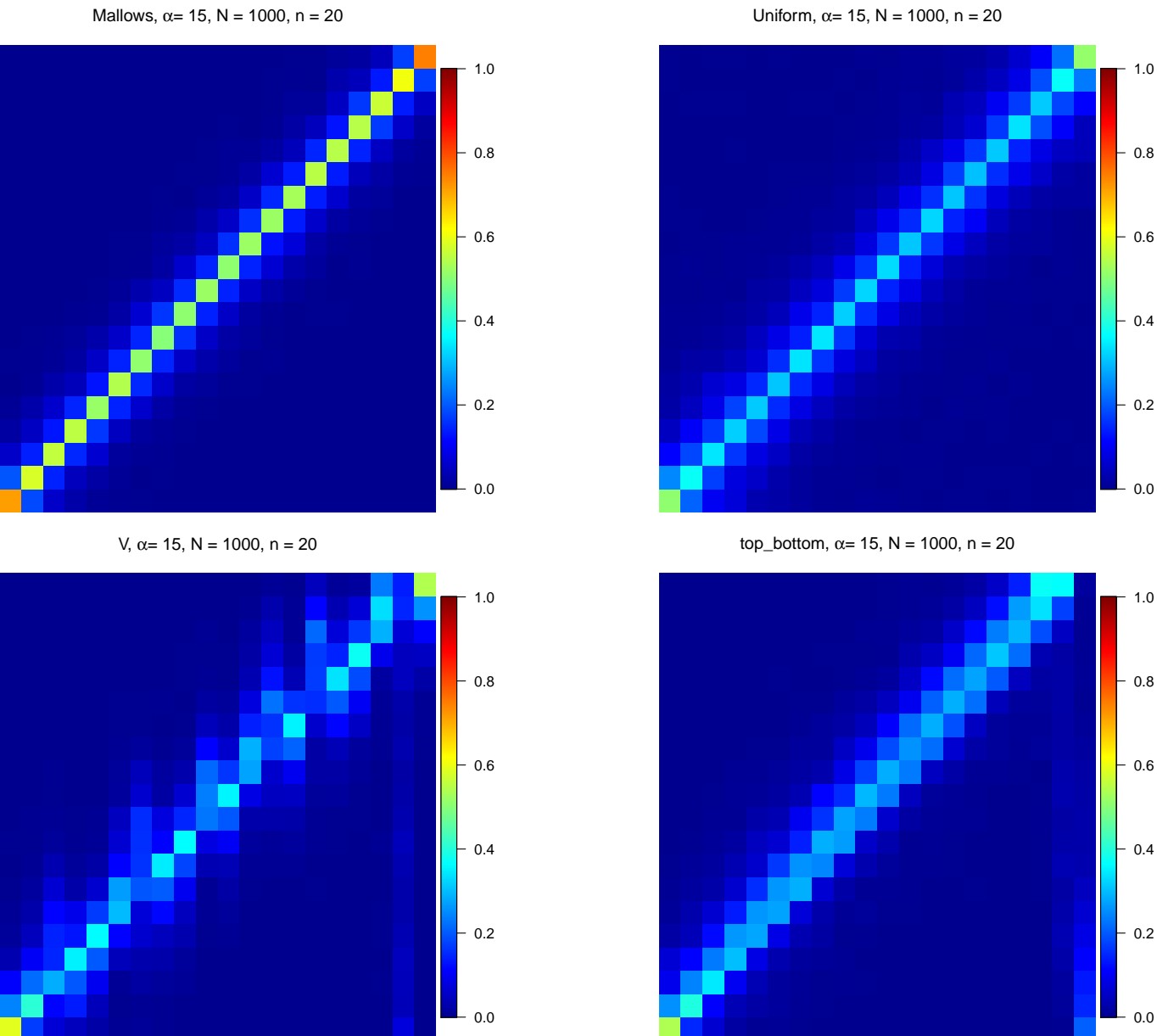

*Figure 10.* Heat plots of $P(R_i^j|\alpha^0, \boldsymbol{\rho}^0)$, when the $\mathbf{R}^j$'s are sampled from the Mallows (top-left), or from the Pseudo-Mallows with different $g(\cdot)$ (top-right: uniform; bottom-left: "V"-rankings; bottom-right: top-bottom ranking). $N = 2000$ samples, $n = 20$ items, $\alpha^0 = 15$.

It can be observed that when individual rankings are drawn from the Mallows distribution, the distribution of $\mathbf{R}^j$ is centred on $\boldsymbol{\rho}^0$, and the same is achieved when using the Pseudo-Mallows with uniform orderings for each sample. If we take a closer look, it can also be discovered that using any choice of the $g(\cdot)$ distribution other than the uniform distribution introduces some biases in the resulting Pseudo-Mallows distribution. As shown in Figure 10, the distribution of the items that tend to be the last sampled ones, is subject to the largest deviation from its position in $\boldsymbol{\rho}^0$. See for example the top- and bottom-ranked items using the "V"-rankings, and the bottom-ranked items using the top-bottom ranking.

A summary of the algorithm for estimating both individual rankings $\{R_1, ..., R_N\}$ and $\boldsymbol{\rho}$ from clicking data is described in Algorithm 5 in Appendix F.

### D.3. Preliminary estimation of $\alpha$ from clicking data

We continue with a similar assumption as in Section 2.4, i.e., that the users' cosine similarity of the given clicking dataset resembles that of a clicking dataset with the same number of items $n$, generated by binarizing full Mallows rankings. As the individual data exists in the form of binary clickings, we define the pairwise user similarity between two binary vectors $\mathbf{B}^j$ and $\mathbf{B}^k$ as

$$\text{sim}(\mathbf{B}^j, \mathbf{B}^k) = \frac{\mathbf{B}^j \cdot \mathbf{B}^k}{|\mathbf{B}^j| \cdot |\mathbf{B}^k|}.$$

More specifically, we first generate full ranking datasets of $N$ users and $n$ items by drawing from the Mallows distribution with a grid of $\alpha^0$ values, and a shared consensus parameter $\boldsymbol{\rho}^0$, as in Section 2.4. These datasets are then binarized by setting, for each user $j$, the top $c_j$ ranked items to 1 and the other items to 0. The number of clicked items $c_j$ for each user $j$ should come from a distribution that best reflects the number of clicks of the given dataset. Possible distributions to consider include the Poisson, the Gamma, and the exponential distributions. Lastly, we calculate the mean pairwise user cosine similarity of each simulated binary dataset, defined as

$$\text{Sim}_{\alpha^0} = \frac{1}{N(N-1)} \sum_{j=1}^{N} \sum_{k \neq j} \text{sim}(\mathbf{B}^j, \mathbf{B}^k). \tag{23}$$

We can then calculate the mean pairwise user cosine similarity of the given dataset, and choose the $\alpha^0$ value of the simulated binary dataset that has the closest mean user cosine similarity.

# E. Additional simulation results

## E.1. Additional results for the simulation study of Section 4.1

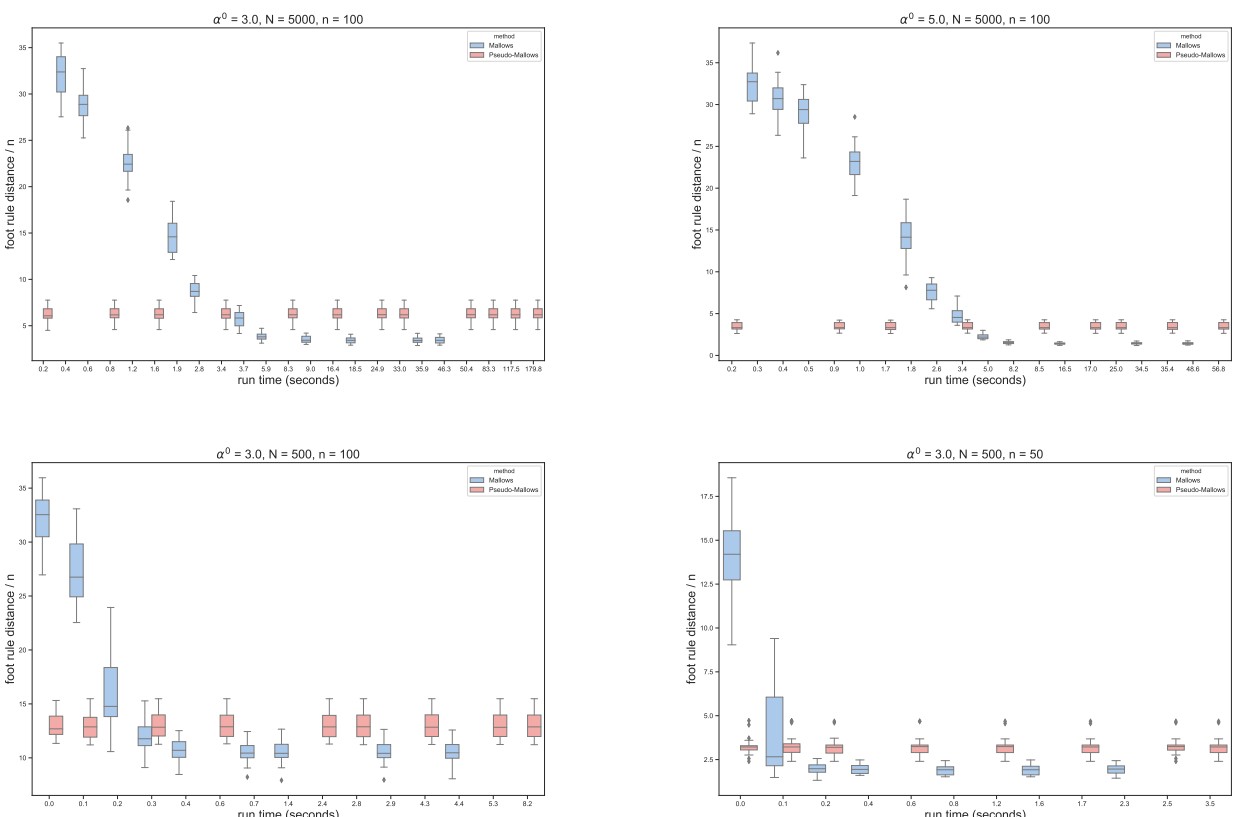

*Figure 11.* Results of the simulation study of Section 4.1: consensus estimation and computation time. $y$-axis: footrule distance between the CP consensus and true consensus $\boldsymbol{\rho}^0$, $x$-axis: computation time in seconds. The $x$-axis is cut some time after convergence of results of both algorithms.

## E.2. Simulation study on individual ranking estimation and personalized recommendations based on clicking data

In this section, we conduct a study to compare the abilities of the Bayesian Mallows MCMC and the Pseudo-Mallows for learning individual rankings and making personalized recommendations.

We first sample full ranking datasets of $N = 500$ users and $n = 50$ items from the Mallows distribution, centred on $\boldsymbol{\rho}^0 = \{1, 2, ..., n\}$, with known $\alpha^0 = 1, 3, 5, 10$. For each $\alpha^0$ value, 20 datasets are generated. The full ranking datasets are then converted to clicking data by deciding on the number of clicked items $c^j$ for each user by sampling from a truncated Poisson distribution with a minimum of 1, a maximum of $n - 3$ and mean 5. The top-$c^j$ ranked items in $\mathbf{R}^j$ are then regarded as clicked for user $j$.

To compare the recommendation accuracy for varying computation time, as in the previous section, for each dataset we run both the Bayesian Mallows MCMC and the Pseudo-Mallows for a grid of different numbers of iterations, leading to different computation times. For the Pseudo-Mallows, the $\mathcal{V}$-set is used to determine the sequence following which each item in the consensus parameter is sampled; while for each user, the uniform distribution is used to decide the sequence following which items are sampled at each iteration, as described in Section 3.1. Based on the samples of the full individual rankings, we calculate the posterior probabilities (11) and make $k = 3$ recommendations for each user. For each user, we denote the set of $k$ recommendation as $Rec^j = \{a_1, ..., a_k\}^j$, and we consider a recommendation $a_i$ as correct if $R^j_{a_i} \in [c_j + 1, c_j + k]$.

The results are displayed in Figure 12, with percentage of correct recommendations made on the $y$-axis against the time it takes to run the algorithms on the $x$-axis. As for the full ranking experiment described in Section 4.1, when not run with

enough iterations for convergence, the Bayesian Mallows MCMC does not produce samples that are truly representative of the target posterior distributions. Having no burn-in period, the Pseudo-Mallows can hence provide effective estimation much faster than the Bayesian Mallows MCMC. If we run the Bayesian Mallows MCMC long enough, eventually, it is able to make individual recommendations that are more accurate than the Pseudo-Mallows, since the data comes from the Mallows distribution. However, when computing time is a constraint, the Pseudo-Mallows estimates of each user's individual full ranking and recommendations are more reliable and precise than the Bayesian Mallows MCMC.

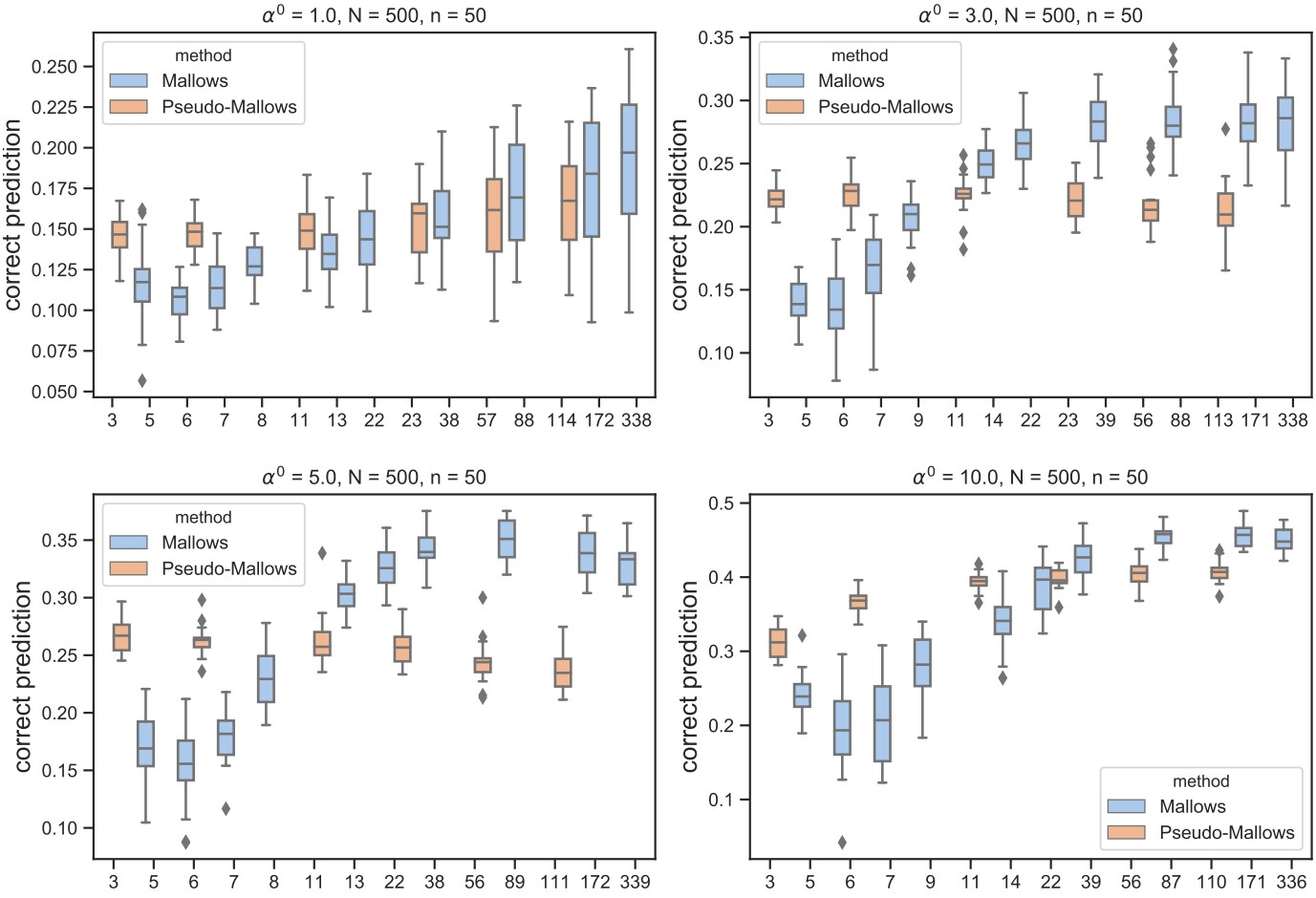

*Figure 12.* Results of the simulation study of Section E.2: recommendation accuracy vs. running time. $y$-axis: percentage of correct recommendations, $x$-axis: computation time in seconds. The $x$-axis is cut some time after convergence of results of both algorithms.

## F. Implementation

A GitHub repository including codes for testing the Pseudo-Mallows algorithm is publicly available (https://github.com/PseudoMallowsAuthors/Pseudo-Mallows). For all Bayesian Mallows MCMC runs reported in this paper we have used the BayesMallows package (Sørensen et al., 2020). All code was run on a regular MacBook.

Pseudo-code schemes of the five algorithms described in the paper are reported below.

---

**Algorithm 1** Pseudo-Mallows Distribution given a fixed ordering

---

**Input:** $n\_samples$, $\alpha^0$, $\{\mathbf{R}^1, ..., \mathbf{R}^N\}$, $\boldsymbol{o}$
**Output:** $\{\boldsymbol{\rho}^1, ..., \boldsymbol{\rho}^{n\_samples}\}$
**for** $t \leftarrow 1$ *to* $n\_samples$ **do**
    set $\boldsymbol{s} = \{1, ..., n\}$    **for** $k \leftarrow 1$ *to* $n$ **do**
        **for** $r \leftarrow 1$ *to* $n$ **do**

$$\text{compute } q_r = \frac{\exp\{-\frac{\alpha^0}{n}\sum_{j=1}^{N} d(R^j_{o_k}, r)\}}{\sum_{z \in \boldsymbol{s}} \exp\{-\frac{\alpha^0}{n}\sum_{j=1}^{N} d(R^j_{o_k}, z)\}} \mathbb{1}_{\{r \in \boldsymbol{s}\}}$$

        **end**
        draw $\rho^t_{o_k} \sim Multinomial(q_1, ..., q_n)$    $\boldsymbol{s} \leftarrow \boldsymbol{s} \setminus \rho^t_{o_k}$
    **end**
**end**

---

**Algorithm 2** Pseudo-Mallows for sampling $\boldsymbol{\rho}$ based on full ranking data

---

**Input:** $n\_samples$, $\alpha^0$, $\{\mathbf{R}^1, ..., \mathbf{R}^N\}$, $\sigma$
**Output:** $\{\boldsymbol{\rho}^1, ..., \boldsymbol{\rho}^{n\_samples}\}$
initialize $\hat{\boldsymbol{\rho}}^0 = \text{rank}(\frac{1}{N}\sum_{j=1}^{N} R^j_1, ..., \frac{1}{N}\sum_{j=1}^{N} R^j_n)$   generate $\mathcal{V}_{\hat{\boldsymbol{\rho}}^0}$
**for** $t \leftarrow 1$ *to* $n\_samples$ **do**
    draw $v^t = \boldsymbol{v}^t$ from $\mathcal{V}_{\hat{\boldsymbol{\rho}}^0}$   **for** $m \leftarrow 1$ *to* $n$ **do**
        draw $v'_m \sim \mathcal{N}(v_m, \sigma)$
    **end**
    compute $v^{t'} = \text{rank}(v'_1, ..., v'_n) \leftrightarrow \{o_1, ..., o_n\}^t$   set $\boldsymbol{s} = \{1, ..., n\}$   **for** $k \leftarrow 1$ *to* $n$ **do**
        **for** $r \leftarrow 1$ *to* $n$ **do**

$$\text{compute } q_r = \frac{\exp\{-\frac{\alpha^0}{n}\sum_{j=1}^{N} d(R^j_{o_k}, r)\}}{\sum_{z \in \boldsymbol{s}} \exp\{-\frac{\alpha^0}{n}\sum_{j=1}^{N} d(R^j_{o_k}, z)\}} \mathbb{1}_{\{r \in \boldsymbol{s}\}}$$

        **end**
        draw $\rho^t_{o_k} \sim Multinomial(q_1, ..., q_n)$    $\boldsymbol{s} \leftarrow \boldsymbol{s} \setminus \rho^t_{o_k}$
    **end**
**end**

---

---

**Algorithm 3** Iterative search algorithm

---

**Input:** $\boldsymbol{v}^0, n\_samples, \alpha^0, \{\mathbf{R}^1, ..., \mathbf{R}^N\}$
**Output:** $\boldsymbol{v}$
$\{\boldsymbol{\rho}^{MCMC}\} \leftarrow$ `BayesMallows::compute_mallows`$(n\_samples, \alpha^0, \{\mathbf{R}^1, ..., \mathbf{R}^N\})$ **for** $l \leftarrow 1$ *to* $max\_iters$ **do**

    **for** $j \leftarrow \{1, ..., n\}$ **do**

        **for** $r \leftarrow \{1, ..., n\}$ **do**

            generate $\{v_1, ..., v_n\}' \leftarrow$ L&S$(\boldsymbol{v}^{l-1}, j, r)$    draw $\{\boldsymbol{\rho}\}^{l,j,r} \leftarrow$ Algorithm 1$(\boldsymbol{v}', \cdot)$   Calculate edge $w_{j,r} = \sum\limits_{i=1}^{n} KL(q(\rho_i^{l,j,r})||p(\rho_i^{MCMC}))$

        **end**

    **end**

    Solve the minimal weight perfect matching $M$ via the Hungarian Method   Compute the new permutation $\boldsymbol{v}^*$ based on $M$   Set $\boldsymbol{v}^l \leftarrow \boldsymbol{v}^*$

**end**

---

**Algorithm 4** Sample for user $j$ based on clicking data

---

**Input:** $\mathbf{B}^j, \alpha, \boldsymbol{\rho}$
**Output:** $\mathbf{R}^j = \{R_1^j, ..., R_n^j\}$
generate $\mathcal{A}^j = \{A_i : B_i^j = 1\}$, $\mathcal{A}^{j^c} = \{A_k : B_k^j = 0\}$   compute $\boldsymbol{\rho}^t = \text{rank}(\rho_i : A_i \in \mathcal{A}^j)$, $\boldsymbol{\rho}^b = \text{rank}(\rho_k : A_k \in \mathcal{A}^{j^c})$
draw ordering $\boldsymbol{o}^t = \{o_1, ..., o_{c_j}\}$ uniformly from $\mathcal{P}_{c_j}$   draw ordering $\boldsymbol{o}^b = \{o_1, ..., o_{c_j^c}\}$ uniformly from $\mathcal{P}_{c_j^c}$   set $\boldsymbol{s}^t = \{1, ..., c_j\}$   **for** *each item $i$ in $\boldsymbol{o}^t$* **do**

    **for** $r \leftarrow 1$ *to* $c_j$ **do**

        compute $q_r = \frac{\exp\{-\frac{\alpha}{n}d(\rho_i^t, r)\}}{\sum\limits_{z \in \boldsymbol{s}^t} \exp\{-\frac{\alpha}{n}d(\rho_i^t, z)\}} \mathbb{1}_{\{r \in \boldsymbol{s}^t\}}$

    **end**

    draw $R_i^j \sim Multinomial(q_1, ..., q_{c_j})$   $\boldsymbol{s}^t \leftarrow \boldsymbol{s}^t \setminus R_i^j$

**end**

set $\boldsymbol{s}^b = \{1, ..., c_j^c\}$   **for** *each item $k$ in $\boldsymbol{o}^b$* **do**

    **for** $r \leftarrow 1$ *to* $c_j^c$ **do**

        compute $q_r = \frac{\exp\{-\frac{\alpha}{n}d(\rho_k^b, r)\}}{\sum\limits_{z \in \boldsymbol{s}^b} \exp\{-\frac{\alpha}{n}d(\rho_k^b, z)\}} \mathbb{1}_{\{r \in \boldsymbol{s}^b\}}$

    **end**

    draw $R_k^j \sim Multinomial(q_1, ..., q_{c_j^c})$   $\boldsymbol{s}^b \leftarrow \boldsymbol{s}^b \setminus R_k^j$

    **for** *each item $k$ in $\boldsymbol{o}^b$* **do**

        $R_k^j \leftarrow R_k^j + c_j$

    **end**

**end**

---

---

**Algorithm 5** Pseudo-Mallows for clicking data

---

**Input:** $\{\mathbf{B}^1, ..., \mathbf{B}^N\}$, $\alpha^0$, $\sigma_{\boldsymbol{\rho}}$, $n\_samples$

**Output:** $\{\mathbf{R}^1, ...., \mathbf{R}^N\}^{1,...,n\_samples}$, $\{\boldsymbol{\rho}^0, ..., \boldsymbol{\rho}^{n\_samples}\}$

initialize $\hat{\boldsymbol{\rho}}^0 = \text{rank}(\frac{1}{N}\sum\limits_{j=1}^{N} B_1^j, ..., \frac{1}{N}\sum\limits_{j=1}^{N} B_n^j)$ **for** $t \leftarrow 1$ *to* $n\_samples$ **do**

> **for** *user* $j \leftarrow 1$ *to* $N$ *in parallel* **do**
> > sample for $\mathbf{R}^{j,t}$ based on $\boldsymbol{\rho}^{t-1}$ using Algorithm 4
>
> **end**
> draw 1 sample for $\boldsymbol{\rho}^t$ based on $\{\mathbf{R}^1, ..., \mathbf{R}^N\}^t$ using Algorithm 2

**end**

---

# G. Proofs

## G.1. Minimization of the expected L-1 distance

*For an item $o_k$, the value of $l^*$ that minimizes the expected value of $\sum\limits_{j=1}^{N} |R_{o_k}^j - l|$ w.r.t. Mallows$(\alpha, \boldsymbol{\rho}^0 | R_{o_k} \neq \rho_{o_1}, ..., \rho_{o_{k-1}}))$,*

*i.e.,*

$$l^* = \underset{l \in \{1,...,n\} \backslash \{\rho_{o_1}, ..., \rho_{o_{k-1}}\}}{\text{argmin}} \mathbb{E}[\sum_{j=1}^{N} |R_{o^k}^j - l|],$$

*is the median of the marginal distribution $R_{o_k}$, given that $R_{o_k} \neq \rho_{o_1}, ..., \rho_{o_{k-1}}$.*

*Proof.*

$$\mathbb{E}[\sum_{j=1}^{N} |R_{o^k}^j - l|]$$

$$= \sum_{j=1}^{N} \mathbb{E}[|R_{o^k}^j - l|]$$

$$= \sum_{j=1}^{N} \sum_{a=1}^{n} |a - l| P_k(a),$$

where $P_k(a) = \sum\limits_{\boldsymbol{r} \in \mathcal{P}_n, r_{o_k} = a, r_{o_k} \neq \rho_{o_1}, ..., \rho_{o_{k-1}}} \frac{1}{Z_n(\alpha)} \exp\{d(\boldsymbol{r}, \boldsymbol{\rho}^0)\}$. The minimization of the expected distance above is

equivalent to minimizing $\sum\limits_{a=1, a \neq \rho_{o_1}, ..., \rho_{o_{k-1}}}^{n} |a - l| P_k(a)$.

$$\sum_{a=1, a \neq \rho_{o_1}, ..., \rho_{o_{k-1}}}^{n} |a - l| P_k(a) = \sum_{a=1, a \neq \rho_{o_1}, ..., \rho_{o_{k-1}}}^{l} (l - a) P_k(a) + \sum_{a=l+1, a \neq \rho_{o_1}, ..., \rho_{o_{k-1}}}^{n} (a - l) P_k(a)$$

$$\frac{d}{dl} \sum_{a=1, a \neq \rho_{o_1}, ..., \rho_{o_{k-1}}}^{n} |a - l| P_k(a) = \sum_{a=1, a \neq \rho_{o_1}, ..., \rho_{o_{k-1}}}^{l} P_k(a) - \sum_{a=l+1, a \neq \rho_{o_1}, ..., \rho_{o_{k-1}}}^{n} P_k(a)$$

$$= 2 \sum_{a=1, a \neq \rho_{o_1}, ..., \rho_{o_{k-1}}}^{l} P_k(a) - 1.$$

Setting the derivative to be 0, we have $\sum\limits_{a=1, a \neq \rho_{o_1}, ..., \rho_{o_{k-1}}}^{l} P_k(a) = \frac{1}{2}$, i.e., $l^*$ is the median of the marginal distribution of

$R_{o_k}$, given $R_{o_k} \neq \rho_{o_1}, ..., \rho_{o_{k-1}}$. $\qquad \square$

## G.2. Symmetry and median of the rank of the middle item

### G.2.1. SYMMETRY OF THE RANK OF THE MIDDLE ITEM

*Given that a ranking $\mathbf{R} \in \mathcal{P}_n$ and $\mathbf{R} \sim Mallows(\boldsymbol{\rho}^0, \alpha)$. Let $o_i^0$ be such that $\rho_{o_i^0}^0 = i$, for $i = 1, ..., n$. For a fixed $\boldsymbol{\rho}^0$ and $\alpha$, $\forall \alpha \in (0, \infty)$, the middle-ranked item in $\boldsymbol{\rho}^0$, i.e., item $o_m^0$ is symmetrically distributed about $m$.*

*Proof.* Recall that the $Mallows(\boldsymbol{\rho}^0, \alpha)$ likelihood function is

$$P(R|\boldsymbol{\rho}^0, \alpha) = \frac{1}{Z_n(\alpha)} \exp\{-\frac{\alpha}{n} d(\mathbf{R}, \boldsymbol{\rho}^0)\}.$$

Given two permutations $\boldsymbol{r}$ and $\boldsymbol{r}' \in \mathcal{P}_n \sim Mallows(\boldsymbol{\rho}^0, \alpha)$, their likelihoods are equal if $d(\boldsymbol{r}, \boldsymbol{\rho}^0) = d(\boldsymbol{r}', \boldsymbol{\rho}^0)$.

For a given $d$, let $\boldsymbol{r}$ be any permutation s.t. $d(\boldsymbol{r}, \boldsymbol{\rho}^0) = d$ and $r_{o_m^0} \neq m$. We define one other permutation $\boldsymbol{r}'$ s.t. $r'_{o_s^0} = 2m - r_{o_{2m-s}^0}$ for $s = 1, ...., 2m - 1$. $\boldsymbol{r}$ and $\boldsymbol{r}'$ have the same distance, i.e. $d(\boldsymbol{r}', \boldsymbol{\rho}^0) = d$, because

$$
\begin{aligned}
d &= \sum_{l=1}^{n} |r_{o_l^0} - \rho_{o_l^0}^0| \\
&= \sum_{l=1}^{n} |r_{o_l^0} - l| \\
&= \sum_{l=1}^{n} |\Delta_l|,
\end{aligned}
\tag{24}
$$

and

$$
\begin{aligned}
d(\boldsymbol{r}', \boldsymbol{\rho}^0) &= \sum_{l=1}^{n} |r'_{o_l^0} - \rho_{o_l^0}^0| \\
&= \sum_{l=1}^{n} |2m - r_{o_{2m-l}^0} - l| \\
&= \sum_{l=1}^{n} |-\Delta_{2m-l}| \\
&= d
\end{aligned}
\tag{25}
$$

This also implies that for the middle item $o_m^0$, and a given $d$, for some $\boldsymbol{r}$ and $k \in \{1, ..., m - 1\}$, if $r_{o_m^0} = m + k$ exists, there is an $\boldsymbol{r}'$ with the same distance and $r'_{o_m^0} = m - k$. Thus it can be concluded that for the middle item $o_m^0$, for all $k = 1, ..., m - 1$, it holds true that

$$P(R_{o_m^0} = m - k|\boldsymbol{\rho}^0, \alpha) = P(R_{o_m^0} = m + k|\boldsymbol{\rho}^0, \alpha),$$

i.e., $R_{o_m^0}$ is symmetrically distributed about $m$. $\qquad\square$

### G.2.2. MEDIAN OF THE RANK OF THE MIDDLE ITEM

*Given that a ranking $\mathbf{R} \in \mathcal{P}_n$ and $\mathbf{R} \sim Mallows(\boldsymbol{\rho}^0, \alpha)$. Let $o_i^0$ be such that $\rho_{o_i^0}^0 = i$, for $i = 1, ..., n$. For a fixed $\boldsymbol{\rho}^0$ and $\alpha$, $\forall \alpha \in (0, \infty)$, for the middle-ranked item in $\boldsymbol{\rho}^0$, i.e., item $o_m^0$, we have*

$$\mathbb{E}[R_{o_m^0}|\boldsymbol{\rho}^0, \alpha] = \rho_{o_m^0}^0 = m \tag{26}$$

*Proof.* For the middle item in $\boldsymbol{\rho}^0$, i.e., item $o_m^0$, as $R_{o_m^0}$ is symmetrically distributed about the value $m$, the median of the distribution is therefore equal to the expected value.

Based on the definition of expected value, we can rewrite (26) as

$$
\begin{aligned}
\mathbb{E}[R_{o_m^0}|\boldsymbol{\rho}^0, \alpha] &= \sum_{\boldsymbol{r} \in \mathcal{P}_n} r_{o_m^0} P(\mathbf{R} = \boldsymbol{r}|\boldsymbol{\rho}^0, \alpha) \\
&= \frac{1}{Z_n(\alpha)} \sum_{\boldsymbol{r} \in \mathcal{P}_n} r_{o_m^0} \exp\left\{ -\frac{\alpha}{n} \boldsymbol{d}(\boldsymbol{r}, \boldsymbol{\rho}^0) \right\} \\
&= \frac{1}{Z_n(\alpha)} \sum_{d \in D} \sum_{\boldsymbol{r}:\boldsymbol{d}(\boldsymbol{r}, \boldsymbol{\rho}^0)=d} r_{o_m^0} \exp\{ -\frac{\alpha}{n} d \};
\end{aligned}
\tag{27}
$$

The normalizing function can be expressed as

$$
Z_n(\alpha) = \sum_{\boldsymbol{r} \in \mathcal{P}_n} \exp\left\{ -\frac{\alpha}{n} \boldsymbol{d}(\boldsymbol{r}, \boldsymbol{\rho}^0) \right\} = \sum_{d \in D} |L_d| \exp\left\{ -\frac{\alpha}{n} \cdot d \right\},
$$

where

$$
L_d = \{\boldsymbol{r} \in \mathcal{P}_n : \boldsymbol{d}(\boldsymbol{r}, \boldsymbol{\rho}^0) = d\},
$$

and $D$ is the set of all possible distances between $\boldsymbol{r}$ and $\boldsymbol{\rho}^0$, $D = \{d_1, ..., d_{|D|}\}$.

We can further express equation (27) as

$$
\begin{aligned}
\mathbb{E}[R_{o_m^0}|\boldsymbol{\rho}^0, \alpha] &= \frac{1}{Z_n(\alpha)} \sum_{d \in D} \sum_{\boldsymbol{r}:\boldsymbol{d}(\boldsymbol{r}, \boldsymbol{\rho}^0)=d} r_{o_m^0} \exp\{ -\frac{\alpha}{n} d \} \\
&= \frac{1}{Z_n(\alpha)} \sum_{d \in D} \sum_{k=1}^{m-1} \left\{ \left[ (m+k)|l_{d,k}^+| \cdot \exp\{ -\frac{\alpha}{n} d \} + (m-k)|l_{d,k}^-| \cdot \exp\{ -\frac{\alpha}{n} d \} \right] \right. \\
&\quad \left. + (m+0)|l_{d,0}| \cdot \exp\{ -\frac{\alpha}{n} d \} \right\};
\end{aligned}
\tag{28}
$$

where

$$
l_{d,k}^+ = \{\boldsymbol{r} \in \mathcal{P}_n : \boldsymbol{d}(\boldsymbol{r}, \boldsymbol{\rho}^0) = d \text{ and } r_{o_m^0} = m + k\}
$$

$$
l_{d,k}^- = \{\boldsymbol{r} \in \mathcal{P}_n : \boldsymbol{d}(\boldsymbol{r}, \boldsymbol{\rho}^0) = d \text{ and } r_{o_m^0} = m - k\}, \text{ and}
$$

$$
l_{d,0} = \{\boldsymbol{r} \in \mathcal{P}_n : \boldsymbol{d}(\boldsymbol{r}, \boldsymbol{\rho}^0) = d \text{ and } r_{o_m^0} = m\}
$$

It holds true that $\sum_{k=1}^{m-1} \left( |l_{d,k}^+| + |l_{d,k}^-| \right) + |l_{d,0}| = |L_d|$

Using the symmetry that we have proven in G.2.1, we can obtain that for the middle item $o_m^0$, and a given $d$, for some $\boldsymbol{r}$ and $k \in \{1, ..., m-1\}$, if $r_{o_m^0} = m + k$ exists, there is an $\boldsymbol{r}'$ with the same distance and $r'_{o_m^0} = m - k$. Therefore, it holds that $|l_{d,k}^+| = |l_{d,k}^-|$, for all $k = 1, ..., m-1$ and for all $d \in D$.

Thus, equation (28) can be further expressed as

$$
\begin{aligned}
\mathbb{E}[R_{o_m^0}|\boldsymbol{\rho}^0, \alpha] =& \frac{1}{Z_n(\alpha)} \sum_{d \in D} \sum_{k=1}^{m-1} \left[ m|l_{d,k}^+| \cdot \exp\{-\frac{\alpha}{n}d\} + m|l_{d,k}^-| \cdot \exp\{-\frac{\alpha}{n}d\}] \right. \\
& \left. + m|l_{d,0}| \cdot \exp\{-\frac{\alpha}{n}d\} \right) \\
=& \frac{1}{Z_n(\alpha)} \sum_{d \in D} \sum_{k=1}^{m-1} \left[ (|l_{d,k}^+| + |l_{d,k}^-|) + |l_{d,0}| \right] \cdot m \cdot \exp\{-\frac{\alpha}{n}d\} \\
=& \frac{1}{Z_n(\alpha)} \sum_{d \in D} |L_d| \cdot m \cdot \exp\{-\frac{\alpha}{n}d\} \\
=& m.
\end{aligned}
\tag{29}
$$

As the median and the expected value are equal, we have that the median of $R_{o_m^0} = m$ □

### G.3. Expectations of two "neighboring" items

Given that a ranking $\mathbf{R} \in \mathcal{P}_n$ and $\mathbf{R} \sim \text{Mallows}(\boldsymbol{\rho}^0, \alpha)$. Let $o_i^0$ be such that $\rho_{o_i^0}^0 = i$, for $i = 1, ..., n$. For a fixed $\boldsymbol{\rho}^0$ and $\alpha$, $\forall \alpha \in (0, \infty)$. For a fixed $j$, for any given $n$ and $\alpha > 0$, we have

$$
\mathbb{E}[R_{o_j^0}|\boldsymbol{\rho}^0, \alpha] < \mathbb{E}[R_{o_{j+1}^0}|\boldsymbol{\rho}^0, \alpha]
$$

.

*Proof.* For a fixed $j$, and for $a < b$, $a \in \{1, 2, ..., n-1\}$, $b \in \{2, ..., n\}$, we first define the set of permutation $\mathcal{F}_{a,b} = \{r : r_{o_j^0} = a, r_{o_{j+1}^0 = b}\}$. Based on this definition, we can partition the space of permutation into six sets:

1. $\mathcal{P}_A = \bigcup_{b=j+2}^{n} \bigcup_{a=j+1}^{b} \mathcal{F}_{a,b}$

2. $\mathcal{P}_{A'} = \bigcup_{b=j+2}^{n} \bigcup_{a=j+1}^{b} \mathcal{F}_{b,a}$

3. $\mathcal{P}_B = \bigcup_{b=1}^{j} \bigcup_{a=1}^{min\{b,j-1\}} \mathcal{F}_{a,b}$

4. $\mathcal{P}_{B'} = \bigcup_{b=1}^{j} \bigcup_{a=1}^{min\{b,j-1\}} \mathcal{F}_{b,a}$

5. $\mathcal{P}_C = \bigcup_{b=j+1}^{n} \bigcup_{a=1}^{j} \mathcal{F}_{a,b}$

6. $\mathcal{P}_{C'} = \bigcup_{b=j+1}^{n} \bigcup_{a=1}^{j} \mathcal{F}_{b,a}$ ,

and it holds true that $\mathcal{P}_n = \mathcal{P}_A \bigcup \mathcal{P}_{A'} \bigcup \mathcal{P}_B \bigcup \mathcal{P}_{B'} \bigcup \mathcal{P}_C \bigcup \mathcal{P}_{C'}$.

We can rewrite the expectations as

$$\mathbb{E}[R_{o_j^0}|\boldsymbol{\rho}^0, \alpha] = \frac{1}{Z_n(\alpha)} \sum_{\boldsymbol{r} \in \mathcal{P}_n} r_{o_j^0} \exp\{-\frac{\alpha}{n} d(\boldsymbol{r}, \boldsymbol{\rho}^0)\}$$

$$= \frac{1}{Z_n(\alpha)} \left[ \sum_{\boldsymbol{r} \in \mathcal{P}_A} r_{o_j^0} \exp\{-\frac{\alpha}{n} d(\boldsymbol{r}, \boldsymbol{\rho}^0)\} + \sum_{\boldsymbol{r} \in \mathcal{P}_{A'}} r_{o_j^0} \exp\{-\frac{\alpha}{n} d(\boldsymbol{r}, \boldsymbol{\rho}^0)\} \right]$$

$$+ \frac{1}{Z_n(\alpha)} \left[ \sum_{\boldsymbol{r} \in \mathcal{P}_B} r_{o_j^0} \exp\{-\frac{\alpha}{n} d(\boldsymbol{r}, \boldsymbol{\rho}^0)\} + \sum_{\boldsymbol{r} \in \mathcal{P}_{B'}} r_{o_j^0} \exp\{-\frac{\alpha}{n} d(\boldsymbol{r}, \boldsymbol{\rho}^0)\} \right] \qquad (30)$$

$$+ \frac{1}{Z_n(\alpha)} \left[ \sum_{\boldsymbol{r} \in \mathcal{P}_C} r_{o_j^0} \exp\{-\frac{\alpha}{n} d(\boldsymbol{r}, \boldsymbol{\rho}^0)\} + \sum_{\boldsymbol{r} \in \mathcal{P}_{C'}} r_{o_j^0} \exp\{-\frac{\alpha}{n} d(\boldsymbol{r}, \boldsymbol{\rho}^0)\} \right],$$

and

$$\mathbb{E}[R_{o_{j+1}^0}|\boldsymbol{\rho}^0, \alpha] = \frac{1}{Z_n(\alpha)} \sum_{\boldsymbol{r} \in \mathcal{P}_n} r_{o_{j+1}^0} \exp\{-\frac{\alpha}{n} d(\boldsymbol{r}, \boldsymbol{\rho}^0)\}$$

$$= \frac{1}{Z_n(\alpha)} \left[ \sum_{\boldsymbol{r} \in \mathcal{P}_A} r_{o_{j+1}^0} \exp\{-\frac{\alpha}{n} d(\boldsymbol{r}, \boldsymbol{\rho}^0)\} + \sum_{\boldsymbol{r} \in \mathcal{P}_{A'}} r_{o_{j+1}^0} \exp\{-\frac{\alpha}{n} d(\boldsymbol{r}, \boldsymbol{\rho}^0)\} \right]$$

$$+ \frac{1}{Z_n(\alpha)} \left[ \sum_{\boldsymbol{r} \in \mathcal{P}_B} r_{o_{j+1}^0} \exp\{-\frac{\alpha}{n} d(\boldsymbol{r}, \boldsymbol{\rho}^0)\} + \sum_{\boldsymbol{r} \in \mathcal{P}_{B'}} r_{o_{j+1}^0} \exp\{-\frac{\alpha}{n} d(\boldsymbol{r}, \boldsymbol{\rho}^0)\} \right] \qquad (31)$$

$$+ \frac{1}{Z_n(\alpha)} \left[ \sum_{\boldsymbol{r} \in \mathcal{P}_C} r_{o_{j+1}^0} \exp\{-\frac{\alpha}{n} d(\boldsymbol{r}, \boldsymbol{\rho}^0)\} + \sum_{\boldsymbol{r} \in \mathcal{P}_{C'}} r_{o_{j+1}^0} \exp\{-\frac{\alpha}{n} d(\boldsymbol{r}, \boldsymbol{\rho}^0)\} \right].$$

Let us first consider $\mathcal{P}_A$ and $\mathcal{P}_{A'}$'s contributions to $\mathbb{E}[R_{o_j^0}|\boldsymbol{\rho}^0, \alpha]$ and $\mathbb{E}[R_{o_{j+1}^0}|\boldsymbol{\rho}^0, \alpha]$. For a a permutation $\boldsymbol{r} \in \mathcal{P}_A$ and its corresponding $\boldsymbol{r}'$ s.t. $r_{o_j^0} = r'_{o_{j+1}^0} = a$, $r_{o_{j+1}^0} = r'_{o_j^0} = b$, and $r_l = r'_l \ \forall i \neq o_j^0, o_{j+1}^0$, it holds that $d(\boldsymbol{r}, \boldsymbol{\rho}^0) = d(\boldsymbol{r}', \boldsymbol{\rho}^0)$ because

$$d(\boldsymbol{r}, \boldsymbol{\rho}^0) = \sum_{i=1}^{j-1} |r_{o_i^0} - \rho_{o_i^0}^0| + |r_{o_j^0} - \rho_{o_j^0}^0| + |r_{o_{j+1}^0} - \rho_{o_{j+1}^0}^0| + \sum_{i=j+2}^{n} |r_{o_i^0} - \rho_{o_i^0}^0|$$

$$= \sum_{i \neq j, j+1} |r_{o_i^0} - i| + |a - j| + |b - (j+1)|$$

$$= \sum_{i \neq j, j+1} |r_{o_i^0} - i| + a - j + b - (j+1),$$

and

$$d(\boldsymbol{r}', \boldsymbol{\rho}^0) = \sum_{i=1}^{j-1} |r'_{o_i^0} - \rho_{o_i^0}^0| + |r'_{o_j^0} - \rho_{o_j^0}^0| + |r'_{o_{j+1}^0} - \rho_{o_{j+1}^0}^0| + \sum_{i=j+2}^{n} |r'_{o_i^0} - \rho_{o_i^0}^0|$$

$$= \sum_{i \neq j, j+1} |r'_{o_i^0} - i| + |b - j| + |a - (j+1)|$$

$$= \sum_{i \neq j, j+1} |r_{o_i^0} - i| + b - j + a - (j+1)$$

$$= d(\boldsymbol{r}, \boldsymbol{\rho}^0)$$

That is to say, for each permutation $\boldsymbol{r} \in \mathcal{P}_A$, we can find another permutation $\boldsymbol{r}' \in \mathcal{P}_{A'}$ with equal distance to $\boldsymbol{\rho}^0$. Therefore, the second row of equation (30) can be derived as

$$
\frac{1}{Z_n(\alpha)} \left[ \sum_{\boldsymbol{r} \in \mathcal{P}_A} r_{o_j^0} \exp\{-\frac{\alpha}{n} d(\boldsymbol{r}, \boldsymbol{\rho^0})\} + \sum_{\boldsymbol{r} \in \mathcal{P}_{A'}} r_{o_j^0} \exp\{-\frac{\alpha}{n} d(\boldsymbol{r}, \boldsymbol{\rho^0})\} \right]
$$

$$
= \frac{1}{Z_n(\alpha)} \left[ \sum_{b=j+1}^{n} \sum_{a=j+1}^{b} \sum_{\boldsymbol{r} \in \mathcal{P}: r_{o_j^0}=a, r_{o_{j+1}^0}=b} a \exp\{-\frac{\alpha}{n} d(\boldsymbol{r}, \boldsymbol{\rho}^0)\} \right.
$$

$$
\left. + \sum_{b=j+1}^{n} \sum_{a=j+1}^{b} \sum_{\boldsymbol{r} \in \mathcal{P}: r_{o_j^0}=b, r_{o_{j+1}^0}=a} b \exp\{-\frac{\alpha}{n} d(\boldsymbol{r}, \boldsymbol{\rho}^0)\} \right]
$$

$$
= \frac{1}{Z_n(\alpha)} \sum_{b=j+2}^{n} \sum_{a=j+1}^{b} (a+b) \sum_{\boldsymbol{r} \in \mathcal{P}: r_{o_j^0}=a, r_{o_{j+1}^0}=b} \exp\{-\frac{\alpha}{n} d(\boldsymbol{r}, \boldsymbol{\rho}^0)\}
$$

$$
= \sum_{b=j+2}^{n} \sum_{a=j+1}^{b} (a+b) P(\mathcal{F}_{a,b}|\boldsymbol{\rho}^0, \alpha)
$$

Similarly for $E[R_{o_{j+1}^o}]$, the second row of equation (31) can be simplified as

$$
\frac{1}{Z_n(\alpha)} \left[ \sum_{\boldsymbol{r} \in \mathcal{P}_A} r_{o_{j+1}^0} \exp\{-\frac{\alpha}{n} d(\boldsymbol{r}, \boldsymbol{\rho^0})\} + \sum_{\boldsymbol{r} \in \mathcal{P}_{A'}} r_{o_{j+1}^0} \exp\{-\frac{\alpha}{n} d(\boldsymbol{r}, \boldsymbol{\rho^0})\} \right]
$$

$$
= \frac{1}{Z_n(\alpha)} \left[ \sum_{b=j+1}^{n} \sum_{a=j+1}^{b} \sum_{\boldsymbol{r} \in \mathcal{P}: r_{o_j^0}=a, r_{o_{j+1}^0}=b} b \exp\{-\frac{\alpha}{n} d(\boldsymbol{r}, \boldsymbol{\rho}^0)\} \right.
$$

$$
\left. + \sum_{b=j+1}^{n} \sum_{a=j+1}^{b} \sum_{\boldsymbol{r} \in \mathcal{P}: r_{o_j^0}=b, r_{o_{j+1}^0}=a} a \exp\{-\frac{\alpha}{n} d(\boldsymbol{r}, \boldsymbol{\rho}^0)\} \right]
$$

$$
= \frac{1}{Z_n(\alpha)} \sum_{b=j+2}^{n} \sum_{a=j+1}^{b} (a+b) \sum_{\boldsymbol{r} \in \mathcal{P}: r_{o_j^0}=a, r_{o_j^0}=b} \exp\{-\frac{\alpha}{n} d(\boldsymbol{r}, \boldsymbol{\rho}^0)\}
$$

$$
= \sum_{b=j+2}^{n} \sum_{a=j+1}^{b} (a+b) P(\mathcal{F}_{a,b}|\boldsymbol{\rho}^0, \alpha)
$$

The second row in (30) and (31) are therefore the same.

Let us now consider $\mathcal{P}_B$ and $\mathcal{P}_{B'}$'s contributions to $\mathbb{E}[R_{o_j^0}|\boldsymbol{\rho}^0, \alpha]$ and $\mathbb{E}[R_{o_{j+1}^0}|\boldsymbol{\rho}^0, \alpha]$.

For any $\boldsymbol{r} \in \mathcal{P}_B$ and its corresponding $\boldsymbol{r}' \in \mathcal{P}_{B'}$ s.t. $r_{o_j^0} = r'_{o_{j+1}^0} = a$, $r_{o_{j+1}^0} = r'_{o_j^0} = b$, and $r_i = r'_i \forall i \neq o_j^0, o_{j+1}^0$, we also have $d(\boldsymbol{r}, \boldsymbol{\rho}^0) = d(\boldsymbol{r}', \boldsymbol{\rho}^0)$, since

$$d(\boldsymbol{r}, \boldsymbol{\rho^0}) = \sum_{i=1}^{j-1} |r_{o_i^0} - \rho_{o_i^0}^0| + |r_{o_j^0} - \rho_{o_j^0}^0| + |r_{o_{j+1}^0} - \rho_{o_{j+1}^0}^0| + \sum_{i=j+2}^{n} |r_{o_i^0} - \rho_{o_i^0}^0|$$
$$= \sum_{i \neq j, j+1} |r_{o_i^0} - i| + |a - j| + |b - (j+1)|$$
$$= \sum_{i \neq j, j+1} |r_{o_i^0} - i| + j - a + (j+1) - b, \text{ and}$$

$$d(\boldsymbol{r'}, \boldsymbol{\rho^0}) = \sum_{i=1}^{j-1} |r'_{o_i^0} - \rho_{o_i^0}^0| + |r'_{o_j^0} - \rho_{o_j^0}^0| + |r'_{o_{j+1}^0} - \rho_{o_{j+1}^0}^0| + \sum_{i=j+2}^{n} |r'_{o_i^0} - \rho_{o_i^0}^0|$$
$$= \sum_{i \neq j, j+1} |r'_{o_i^0} - i| + |b - j| + |a - (j+1)|$$
$$= \sum_{i \neq j, j+1} |r_{o_i^0} - i| + j - b + (j+1) - a$$
$$= d(\boldsymbol{r}, \boldsymbol{\rho^0})$$

Following similar logic, we can obtain that the third row in (30) and (31) are also the same.

Now let us consider $\mathcal{P}_C$ and $\mathcal{P}_{C'}$'s contributions to $\mathbb{E}[R_{o_j^0}]$ and $\mathbb{E}[R_{o_{j+1}^0}]$.

For any given permutation $\boldsymbol{r} \in \mathcal{P}_C$ and its corresponding $r' \in \mathcal{P}_{C'}$ s.t. $r_{o_j^0} = r'_{o_{j+1}^0} = a$, $r_{o_{j+1}^0} = r'_{o_j^0} = b$, and $r_l = r'_l$ $\forall i \neq o_j^0, o_{j+1}^0$ , we have

$$d(\boldsymbol{r}, \boldsymbol{\rho^0}) = \sum_{i=1}^{j-1} |r_{o_i^0} - \rho_{o_i^0}^0| + |r_{o_j^0} - \rho_{o_j^0}^0| + |r_{o_{j+1}^0} - \rho_{o_{j+1}^0}^0| + \sum_{i=j+2}^{n} |r_{o_i^0} - \rho_{o_i^0}^0|$$
$$= \sum_{i \neq j, j+1} |r_{o_i^0} - i| + |a - j| + |b - (j+1)|$$
$$= \sum_{i \neq j, j+1} |r_{o_i^0} - i| + j - a + b - (j+1)$$
$$= \sum_{i \neq j, j+1} |r_{o_i^0} - i| + b - a - 1, \text{ and}$$

$$d(\boldsymbol{r'}, \boldsymbol{\rho^0}) = \sum_{i=1}^{j-1} |r'_{o_i^0} - \rho_{o_i^0}^0| + |r'_{o_j^0} - \rho_{o_j^0}^0| + |r'_{o_{j+1}^0} - \rho_{o_{j+1}^0}^0| + \sum_{i=j+2}^{n} |r'_{o_i^0} - \rho_{o_i^0}^0|$$
$$= \sum_{i \neq j, j+1} |r'_{o_i^0} - i| + |b - j| + |a - (j+1)|$$
$$= \sum_{i \neq j, j+1} |r_{o_i^0} - i| + b - j + (j+1) - a$$
$$= \sum_{i \neq j, j+1} |r_{o_i^0} - i| + b - a + 1$$
$$= d(\boldsymbol{r}, \boldsymbol{\rho^0}) + 2.$$

The last row of (30) can be derived as

$$\frac{1}{Z_n(\alpha)}\left[\sum_{\boldsymbol{r}\in\mathcal{P}_C} r_{o_j^0}\exp\{-\frac{\alpha}{n}d(\boldsymbol{r},\boldsymbol{\rho^0})\} + \sum_{\boldsymbol{r}\in\mathcal{P}_{C'}} r_{o_j^0}\exp\{-\frac{\alpha}{n}d(\boldsymbol{r},\boldsymbol{\rho^0})\}\right]$$

$$=\frac{1}{Z_n(\alpha)}\left[\sum_{b=j+1}^{n}\sum_{a=1}^{j}\sum_{\boldsymbol{r}\in\mathcal{P}:r_{o_j^0}=a,r_{o_{j+1}^0}=b} a\exp\{-\frac{\alpha}{n}d(\boldsymbol{r},\boldsymbol{\rho^0})\}\right.$$

$$\left.+\sum_{b=j+1}^{n}\sum_{a=1}^{j}\sum_{\boldsymbol{r}\in\mathcal{P}:r_{o_j^0}=b,r_{o_{j+1}^0}=a} b\exp\{-\frac{\alpha}{n}d(\boldsymbol{r},\boldsymbol{\rho^0})\}\right]$$

$$=\frac{1}{Z_n(\alpha)}\sum_{b=j+1}^{n}\sum_{a=1}^{j}\sum_{\boldsymbol{r}\in\mathcal{P}:r_{o_j^0}=b,r_{o_{j+1}^0}=a} a\exp\{-\frac{\alpha}{n}d(\boldsymbol{r},\boldsymbol{\rho^0})\} + b\exp\{-\frac{2\alpha}{n}\}\exp\{-\frac{\alpha}{n}d(\boldsymbol{r},\boldsymbol{\rho^0})\}$$

$$=\sum_{b=j+1}^{n}\sum_{a=1}^{j}(a+b\exp\{-\frac{2\alpha}{n}\})P(\mathcal{F}_{a,b}|\boldsymbol{\rho^0},\alpha).$$

Similarly, we can obtain that the last row of (31) is

$$\frac{1}{Z_n(\alpha)}\left[\sum_{\boldsymbol{r}\in\mathcal{P}_C} r_{o_{j+1}^0}\exp\{-\frac{\alpha}{n}d(\boldsymbol{r},\boldsymbol{\rho^0})\} + \sum_{\boldsymbol{r}\in\mathcal{P}_{C'}} r_{o_{j+1}^0}\exp\{-\frac{\alpha}{n}d(\boldsymbol{r},\boldsymbol{\rho^0})\}\right]$$

$$=\sum_{b=j+1}^{n}\sum_{a=1}^{j}(b+a\exp\{-\frac{2\alpha}{n}\})P(\mathcal{F}_{a,b}|\boldsymbol{\rho^0},\alpha).$$

Hence, the difference between $\mathbb{E}[R_{o_{j+1}^0}|\boldsymbol{\rho^0},\alpha]$ and $\mathbb{E}[R_{o_j^0}|\boldsymbol{\rho^0},\alpha]$ reduces to:

$$\mathbb{E}[R_{o_{j+1}^0}|\boldsymbol{\rho^0},\alpha] - \mathbb{E}[R_{o_j^0}|\boldsymbol{\rho^0},\alpha] = \sum_{b=j+1}^{n}\sum_{a=1}^{j}(b-a)(1-\exp\{-\frac{2\alpha}{n}\})P(\mathcal{F}_{a,b}|\boldsymbol{\rho^0},\alpha).$$

It can be easily observed that every term in the above equation are positive for all $\alpha > 0$, and we can conclude that $\mathbb{E}[R_{o_{j+1}^0}|\boldsymbol{\rho^0},\alpha] > \mathbb{E}[R_{o_j^0}|\boldsymbol{\rho^0},\alpha]$ for all $\alpha > 0$.

$\square$

