# OpenReview forum: "Pseudo-Mallows for Efficient Probabilistic Preference Learning"
_ICML.cc/2026/Conference — ICML 2026 regular_

### Official Review · Reviewer_1UEK · 2026-03-11

**Soundness:** 4
**Presentation:** 3
**Significance:** 4
**Originality:** 4
**Overall Recommendation:** 5
**Confidence:** 3

**Summary:**

The Bayesian Mallows model is a probabilistic model used to represent user preferences in recommender systems. These models have an intractable posterior distribution and use MCMC or augmentation to infer it. However, these methods have high running times and are therefore not suitable for large datasets, which often occur in real-life recommender systems. The authors propose the first variational inference method to infer this posterior. In particular, they propose a pseudo-mallows model and show how to use it to learn from full ranking data and partial ranking data. Simulation results using a ground truth Mallows model show slightly worse performance but a much faster running time. Performance on a real-life dataset shows better performance due to MCMC not converging in the time limit.

**Compliance With Llm Reviewing Policy:**

Affirmed.

**Final Justification:**

Clearly a good paper, and the rebuttals suggest that the final version will address the remaining issues raised by the reviewers.

**Key Questions For Authors:**

-

**Limitations:**

Yes

**Strengths And Weaknesses:**

The paper is well written and introduces the running time limitation of classic MCMC methods for Bayesian Mallows model as a problem worthwhile solving. They are the first to propose a variational inference method for this problem and are therefore, by definition, original. The authors provide enough details to replicate the method. The proposal of their Pseudo-Mallows model is convincing and the results show high promise.

The choice of boxplots in Figure 1 does not make sense. The x-axis is a continuous variable. Instead, using a line with a confidence interval around it would make the plot much easier to read.

Due to the page limit, a lot of the information of the paper is in the appendix, at times making it hard to read the paper without looking at the appendix.

The proposed method’s selling point is its fast running time, allowing it to be used on large datasets. The simulation study, on the other hand, only covers relatively small datasets where running times are not an issue. It would be interesting to see whether the results remain similar for very large datasets.

---

> ### Author Rebuttal · Authors · 2026-03-31
>
> Thank you for acknowledging the novelty and strengths of our paper.
>
> We agree with your comment on Figure 1 and will fix this in the revised version of the paper.
>
> We will bring forward to the main paper more information from the appendix, making it easier to read the paper. The extra page allowed for the final version will allow to do this.
>
> The extra page will also allow us to include an additional study on a larger datasets, confirming that the advantage of using Pseudo-Mallows is even bigger then, since the MCMC in BayesMallows scales quadratically in $n$, while the Pseudo-Mallows is insensitive to $n$.

---

> > ### Author Rebuttal · Reviewer_1UEK · 2026-04-01
> >
> > Thank you for your response, I have no further questions.

---

### Official Review · Reviewer_4Prq · 2026-03-13

**Soundness:** 2
**Presentation:** 1
**Significance:** 1
**Originality:** 3
**Overall Recommendation:** 4
**Confidence:** 3

**Summary:**

The paper is about ranking estimation. In this context, n objects are ranked (fully or partially) by N individuals, and one wants to reconstruct a latent consensus ranking and all the individual ones, if those were not fully observed. The authors propose Pseudo-Mallows models as a factorization of the Mallows likelihood that compares sequentially each entry of the individual rankings with the latent consensus ones, conditionally on the ordering of such sequential decomposition. The resulting posterior is presented as a variational approximation of the Bayesian Mallows posterior. The variational approach is formulated as optimization over all the possible orderings of the conditional Pseudo-Mallows posterior, and its extension to the full proposed posterior is based on a conjecture. Simulations form Mallows models are included to show that the MCMC approach stabilizes, after convergence, around the same accuracy as the Pseudo-Mallows. The two methods are also compared on real data, with fully and partially observed rankings.

**Compliance With Llm Reviewing Policy:**

Affirmed.

**Final Justification:**

The authors have addressed the majority of my concerns.
I believe the paper presents an interesting idea which, with some notational refinements and the addition of a numerical study on larger datasets that clearly demonstrates its advantages, would be well supported by convincing empirical results. The method would still lack a clear theoretical contribution, but I understand that a full proof may be difficult to obtain.
I am happy to raise my score to 4.

**Key Questions For Authors:**

1. In Section 2.2. the ordering “corresponding to a ranking” is only implicitly defined in Definition 2.1, contextually with the V-rankings. Wouldn’t an inverse permutation notation make more clear the relationship between these objects?
2. The V-rankings are defined in relation to a ranking, not to its ordering, yet the notation $V_{o^0}$ is later used, then followed by $V_{\rho^0}$
3. Rankings and orderings are alternatively linked to each other (as in $\rho^0$ and $o^0$) or not (as in $\rho$ and $o$ in the conjecture, or in the original formulation of the model), without a change in notation that underlines it.
4. The sentence “the optimal choice for $g(i)$ is a $\delta$-distribution with its density concentrated uniformly on all V-rankings that belong to $V_{\rho_0}$” is unclear.
5. What is the practical advantage of passing to the “full model” by choosing a distribution for i (or o)?

**Limitations:**

yes

**Strengths And Weaknesses:**

Soundness.
The extension of the variational framework to the full model is based on a conjecture whose intuition is given in appendix with a very heuristic explanation. The effectiveness of the method as approximation of the Bayesian Mallows is not properly sustained by theoretical results, nor by extensive empirical ones. The overall performance in the broader context of recommender systems is not really established, since it is only compared to the Bayesian Mallows.

Presentation.
Even if the purpose of the paper is clear, it results generally difficult to read. Sections 2.2 is particularly unclear, with undefined objects, several confused sentences and clashing notations (see key questions). This does not help understanding the nature of the conjecture on which the variational argument is based, nor its soundness.

Significance.
The method is presented as a variational approximation of a used ranking model, but it lacks theoretical validation. Simulations are showing that MCMC picks up quickly with the proposed method’s accuracy, so the motivation is not showed to be strong, and no direct measure of posterior approximation are given. On the other hand Pseudo-Mallows is not really validated as a stand-alone ranking estimation method (it is only compared to Bayesian Mallows on real data).

Originality.
As far as I know, and as the authors declare, the proposal of a VI approximation for a ranking estimation model, or in general in the context of inference over the space of permutations, is novel. This original idea is not sufficiently developed in this paper, though.

---

> ### Author Rebuttal · Authors · 2026-03-31
>
> We thank the reviewer for the constructive comments. As for the soundness of the main theoretical result, the Appendix reports several arguments in support of the conjecture, both empirical (Appendix C2) and theoretical (Appendix C1, supported by proofs of relevant results in Appendix H), and a study on finite-sample estimation (Appendix D). Some explanations are heuristic, but some theoretical results are also there, and the empirical exploration is quite extensive. We can bring important parts of this into the main paper thanks to the additional extra space. The clarity of the presentation can also be improved and will help to clarify the soundness of our conjecture. We acknowledge that our conjecture, despite rather strong supportive evidence, remains a conjecture. A well supported conjecture like ours will require significant work to be proved in some generality, and we leave this to the mathematical community and future work. For practical purposes, we believe that we have good arguments to using our variational approximation instead of Bayes Mallows.
>
> As for the significance of our proposal, we did not really validate Pseudo-Mallows as a stand-alone ranking estimation method because such validation has already been carried out extensively for the Bayes Mallows (see Liu et al., 2019a and Liu et al., 2019b), both with respect to standard matrix factorization approaches such as collaborative filtering, and to other probabilistic frameworks such as Plackett-Luce or Bradley-Terry. We therefore focused on assessing the performance of Pseudo-Mallows as an approximation to the Bayes Mallows MCMC-based estimation.
>
> It is true that MCMC picks up quickly with the proposed method’s accuracy, and indeed Figure 1 was chosen precisely as the advantage of the Pseudo-Mallows is visible on the same order of computing time as for the MCMC-based BayesMallows, which is due to $n$ not being too large. In realistic applications, $n$ would be up to 3 orders of magnitude larger than in Figure 1, and since the BayesMallows scales quadratically in $n$, while the Pseudo-Mallows is insensitive to $n$, BayesMallows would run in days while Pseudo-Mallows still in seconds. Since the Pseudo-Mallows provides an approximated solution to the true posterior, while MCMC sampling is exact, obviously BayesMallows will outperform the Pseudo-Mallows in the long run, but this gain in accuracy will happen in huge computing times. Therefore, showing in Figure 1 the accuracy gain achievable in much shorter time via Pseudo-Mallows gives a strong motivation for such accuracy gain to be even more critical for larger settings. In a revision, we will add a study on a much larger dataset, showing clearly that the improvement in computing time of using Pseudo-Mallows instead of BayesMallows is substantial.
>
> As for the originality, we agree that we could emphasize better the important novelty of our idea, and will try to do that in a revised version.
>
> Point-to-point rebuttal to the key questions for authors:
>
> 1. Yes, we can change that. The implicit definition was used to have a compact mathematical notation, to save some space.
>
> 2. We can update the notation accordingly.
>
> 3. We can update the notation to make this more coherent.
>
> 4. We can clarify this sentence; any of the V-rankings in the $V$-set are equivalent in terms of contributions to the ELBO and therefore a uniform distribution over the $V$-set is optimal.
>
> 5. By definition of the $V_{\rho_0}$-set, V-rankings are not unique. From the empirical study in Appendix C, it can be observed that all V-rankings appear to be equivalently effective at achieving a good approximation of the Mallows posterior. The practical advantage of passing to the “full model” is hence that by choosing a distribution for i (or o) you allow the algorithm to explore using not only one V-ranking, but all, which are equally effective, hence making better inference. We should definitely make this point clearer in the paper.
>
> References
>
> Liu, Q., Reiner, A. H., Frigessi, A., & Scheel, I. (2019a). Diverse personalized recommendations with uncertainty from implicit preference data with the Bayesian Mallows model. Knowledge-Based Systems, 186, 104960.
>
> Liu, Q., Crispino, M., Scheel, I., Vitelli, V., & Frigessi, A. (2019b). Model-based learning from preference data. Annual review of statistics and its application, 6(1), 329-354.

---

> > ### Author Rebuttal · Reviewer_4Prq · 2026-04-02
> >
> > The authors have addressed the majority of my concerns.
> > I believe the paper presents an interesting idea which, with some notational refinements and the addition of a numerical study on larger datasets that clearly demonstrates its advantages, would be well supported by convincing empirical results. The method would still lack a clear theoretical contribution, but I understand that a full proof may be difficult to obtain.
> > I am happy to raise my score to 4.

---

> > > ### Author Response · Authors · 2026-04-02
> > >
> > > Thank you! And we will follow your indications in preparing the paper for publication.

---

### Official Review · Reviewer_DE5i · 2026-03-16

**Soundness:** 3
**Presentation:** 3
**Significance:** 3
**Originality:** 3
**Overall Recommendation:** 4
**Confidence:** 3

**Summary:**

The paper proposes the Pseudo-Mallows distribution, as an approximation of the posterior distribution of the Bayesian Mallows model.  The purpose is tractability as the Bayesian Mallows model has complex inference.  As an instance, the model is fit to clicking data, and is tested on a personal recommendation application, showing that indeed it allows faster probabilistic preference learning.

**Compliance With Llm Reviewing Policy:**

Affirmed.

**Key Questions For Authors:**

Q1. Why is the experiment conducted only on one dataset?

Q2. Why not include a comparison to standard recommendation algorithms, such as BPR?

**Strengths And Weaknesses:**

Strengths:

S1. Clear motivation for the need for approximating the bayesian mallows for recommendation applications.

S2. Good formalization of the pseudo-mallows distribution, with optimality conjecture, as well as inference analysis.

S3. Experiments show gains in computational speed and accuracy.


Weaknesses:

W1. Experiments only on one dataset, Norwegian Broadcasting Corporation (NRK), which is not a common benchmark dataset for recommendations.

W2. Some comparison to standard recommendation algorithms such as BPR based on pairwise ranking may be useful.

---

> ### Author Rebuttal · Authors · 2026-03-31
>
> We are grateful to the reviewer for acknowledging important strengths of our paper. Below we address the two weaknesses pointed out by the reviewer.
>
> W1. We thought it would be interesting to show results on a non-standard benchmark example such as the NRK data. We have done a simulation study comparing the abilities of Pseudo-Mallows and BayesianMallows  to learn individual rankings and make personalized recommendations from clicking data, that confirms that the Pseudo-Mallows can provide effective estimation much faster than the BayesianMallows. We will add this to the paper, in part in the supplementary material.  Alternatively, we can run a comparison on a second benchmark dataset, possibly the FINN.no slate data set (https://opensource.finntech.no/recsys_slates_dataset/; Eide, Leslie & Frigessi, 2022).
>
> W2. The scope of this paper is not really to evaluate Pseudo-Mallows as a recommendation algorithm, but to evaluate its performance as an approximation to the intractable posterior distribution of the Bayesian Mallows model (BMM). A comparison of the performance of BMM to other standard probabilistic recommendation algorithms was already carried out. Liu et al. (2019a) presents a comparison between BMM and state-of-the-art matrix factorization method (CF) to make recommendations from clicking data, showing that BMM makes personalized recommendations with similar accuracy as CF, while also achieving much higher level of diversity, and producing interpretable uncertainty estimation. The review paper Liu et al. (2019b) compares BMM to other probabilistic approaches to preference learning, including the widely used Plackett–Luce and Bradley–Terry models. If, as we show, Pseudo-Mallows is a good approximation of BMM, it is still possible that the approximation blurs te advantages that BMM has on competing models. Therefore, we will try to add in a revised version of the paper, some comparisons.
>
> References
>
> Eide, S., Leslie, D. S., & Frigessi, A. (2022). Dynamic slate recommendation with gated recurrent units and Thompson sampling. Data Mining and Knowledge Discovery, 36(5), 1756-1786.
>
> Liu, Q., Reiner, A. H., Frigessi, A., & Scheel, I. (2019a). Diverse personalized recommendations with uncertainty from implicit preference data with the Bayesian Mallows model. Knowledge-Based Systems, 186, 104960.
>
> Liu, Q., Crispino, M., Scheel, I., Vitelli, V., & Frigessi, A. (2019b). Model-based learning from preference data. Annual review of statistics and its application, 6(1), 329-354.

---

> > ### Author Rebuttal · Reviewer_DE5i · 2026-04-03
> >
> > Thank you for providing a rebuttal. The authors have mostly addressed the earlier points raised.

---

> > > ### Author Response · Authors · 2026-04-03
> > >
> > > We are glad to note that we have sufficiently addressed your concerns. We hope these clarifications may justify a positive re-evaluation.

---

### Official Review · Reviewer_Qsx6 · 2026-03-24

**Soundness:** 2
**Presentation:** 2
**Significance:** 3
**Originality:** 3
**Overall Recommendation:** 4
**Confidence:** 3

**Summary:**

This paper introduces the Pseudo-Mallows distribution as a variational inference approximation to the Bayesian Mallows model posterior for probabilistic preference learning. The Pseudo-Mallows is constructed as a product of univariate conditional distributions over permutation elements, where the factorization ordering serves as the variational parameter. The authors conjecture that the optimal ordering follows a "V-set" structure based on the consensus ranking, support this with empirical evidence and partial theory, and extend the method to handle clicking data for personalized recommendations. Experiments on simulated and real (NRK) data show substantial speed advantages over MCMC-based inference.

**Compliance With Llm Reviewing Policy:**

Affirmed.

**Final Justification:**

The rebuttal addressed my concerns about the conjecture.

**Key Questions For Authors:**

Same as weaknesses.

**Limitations:**

Yes

**Strengths And Weaknesses:**

**Strengths:**
1. The core idea of constructing a tractable variational family over permutations that avoids the standard mean-field assumption is well-motivated. The Pseudo-Mallows distribution retains the numerator structure of the Mallows model while enabling independent sampling.

**Weaknesses:**
1. The central theoretical claim — the Optimal Pseudo-Mallows Conjecture — remains unproven. While the empirical evidence is supportive, the paper relies heavily on this conjecture for its main algorithmic contribution. The asymptotic nature of the conjecture (requiring N → ∞) further limits its theoretical applicability to finite-sample settings. The authors should discuss more explicitly what could go wrong if the conjecture fails in certain regimes, and whether any weaker formal guarantees can be established.
2. The estimation of the scale parameter α via cosine similarity matching (Section 2.5) is somewhat ad hoc. It requires simulating datasets on a grid of α values and finding the closest match, which introduces an additional source of approximation error that is not carefully characterized. There is no analysis of how sensitive the downstream inference is to errors in α estimation.
3. The paper's presentation could be improved. The main text is dense with notation, and the flow between the full-ranking case (Section 2) and the clicking data extension (Section 3) is abrupt. Key algorithmic details are deferred to Appendix G, making it difficult to understand the complete method from the main text alone. The paper would benefit from a clearer high-level algorithmic summary earlier on.
4. The experimental comparisons are limited to MCMC as the sole baseline. Other scalable approaches for preference learning or rank aggregation — such as spectral methods, EM-based approaches, or other VI methods for discrete distributions — are not considered. This makes it difficult to assess where the Pseudo-Mallows stands in the broader landscape of efficient preference learning methods.

---

> ### Author Rebuttal · Authors · 2026-03-31
>
> We thank the reviewer for acknowledging our new approach to the variational approximation of the Bayes Mallows posterior, that goes beyond the mean field construction while still retaining the Mallows numerator structure.
>
> We acknowledge that the theoretical details can be heavy and tightly packed (like in Section 2 to 3), but this was due to space constraints. In a revised version of the paper, we will simplify the notation as much as possible and improve the presentation of the theory. We will also add a clear high-level algorithmic summary at the end of each theoretical Section.
>
> Concerning our conjecture on the optimality of the $\mathcal{V}$-set, we explored a robustified version of the conjecture suitable to finite-sample settings in Section 2.4.2, and we added ample supportive empirical evidence to the paper (see Section D in the Supplementary materials). We will extend these discussions in a revised version of the paper by adding an exploration of the $\mathcal{V}$-set behavior under regimes when the conjecture fails, to add insight and concreteness and possibly weaker formal guarantees. We do not think that it is possible yet to prove our conjecture under wide assumptions, despite rather convincing evidence, and we leave it to further work, like often happens with mathematical conjectures, whose formulation is an important contribution per se.
>
> For what concerns the estimation of the scale parameter $\alpha$, we are aware that this introduces further posterior approximation. However, the sensitivity of downstream inference to the choice of this parameter has been previously explored in the context of Bayes Mallows (see Table 3 in Eliseussen, Fleischer & Vitelli, 2022), providing evidence that the sensitivity is limited. The limited sensitivity is because $\alpha$ is scalar and its slight misspecification only influences the overall dispersion of the Mallows posterior, but not the respective ranking of the items in either the consensus or the latent individual rankings.
>
> Experimental comparisons with alternative methods. We compare our approach to the MCMC-based BayesMallows because this is the gold standard for fitting the Bayesian Mallows distribution, and hence the method Pseudo-Mallows is meant to improve on. The Bayesian Mallows distribution is capable of both quantifying the uncertainty for estimates, and of individual-level prediction beside rank aggregation, as required for rank-based recommendation tasks such as the NRK prediction task. We do not know of any alternative methods that are capable of this. Spectral methods for recommendations do not quantify the uncertainty in estimates (Liu, Yu & Bay, 2024), EM-based methods are slow (see the extensive comparisons in Eliseussen, Frigessi & Vitelli, 2023). As far as we know, the only comparable variational approximations for discrete data are tailored to Bipartite Matching (Volkovs & Zemel, 2012) or developed to deal with biological sequences (Bouchard-Coté & Jordan, 2010; Rao & Kirk, 2025), thus being suitable for a different class of problems. We will write these points clearly in the next version of our paper.
>
>
>
> References
>
> Eliseussen, E., Fleischer, T., & Vitelli, V. (2022). Rank‐based Bayesian variable selection for genome‐wide transcriptomic analyses. Statistics in Medicine, 41(23), 4532-4553.
>
> Liu, Q., Yu, M., & Bai, M. (2024). A study on a recommendation algorithm based on spectral clustering and GRU. Iscience, 27(2).
>
> Eliseussen, E., Frigessi, A., & Vitelli, V. (2023). Rank-based Bayesian clustering via covariate-informed Mallows mixtures. arXiv preprint arXiv:2312.12966.
>
> Volkovs, M., & Zemel, R. (2012). Efficient sampling for bipartite matching problems. Advances in Neural Information Processing Systems, 25.
>
> Bouchard-Côté, A., & Jordan, M. (2010). Variational inference over combinatorial spaces. Advances in Neural Information Processing Systems, 23.
>
> Rao, J., & Kirk, P. D. (2025). VICatMix: variational Bayesian clustering and variable selection for discrete biomedical data. Bioinformatics Advances, 5(1), vbaf055.

---

> > ### Author Rebuttal · Reviewer_Qsx6 · 2026-04-05
> >
> > Thank you for the response. My concerns are fully resolved and I have adjusted my score.

---

> > > ### Author Response · Authors · 2026-04-05
> > >
> > > Thank you! We are happy that your concerns have been fully resolved.

---

### Decision · Program_Chairs · 2026-04-30

**Decision:**

Accept (regular)

**Comment:**

In this paper, the authors propose the Pseudo-mallows distribution as a VI approximation to the BMM and develops a VI method for a discrete distribution over permutations. The authors conjecture the optimal permutation in the family of the Pseudo Mallows distribution

All the reviewers are positive about the elegant idea - indeed, the VI method makes inference significantly faster. However, some shared concerns are raised as well. Primarily, the key concern is the theoretical validity of the conjecture regarding the optimal permutation - more evidence in the form of its optimality proof in certain special conditions or numerical results for small values of $n$ can make this paper stronger. This is because the entire paper rests on the conjecture and overall empirical results are also limited.

Reviewers have also raised concerns about the paper being hard to read with dense notations. The authors should improve the notations and the presentation in the camera-ready.